



# Arctic (Svalbard islands) active and exported diatom stocks and cell health status

**Susana Agustí[1], Jeffrey W. Krause[2,3], Israel A. Marquez[2,3], Paul Wassmann[4], Svein Kristiansen[4], and Carlos M. Duarte[1,5]**

[1]King Abdullah University of Science and Technology, Thuwal, 23955-6900, Kingdom of Saudi Arabia TS1
[2]Dauphin Island Sea Lab, Dauphin Island, AL 36528-4603, USA TS2
[3]Department of Marine Sciences, University of South Alabama, Mobile, AL CE1 36688-0002, USA
[4]Department of Arctic and Marine Biology, UiT The Arctic University of Norway, 9037 Tromsø, Norway
[5]Arctic Research Centre, Department of CE2 Bioscience, Aarhus University, C.F. Mollers Alle 8, 8000 Aarhus C, Denmark

**Correspondence:** Susana Agustí (susana.agusti@kaust.edu.sa)

**Abstract.** Diatoms CE3 tend to dominate the Arctic spring phytoplankton bloom, a key event in the ecosystem including a rapid decline in surface-water $p\mathrm{CO}_2$. While a mass sedimentation event of diatoms at the bloom terminus is commonly observed, there are few reports on the status of diatoms' health during Arctic blooms and its possible role on sedimentary fluxes. Thus, we examine the idea that the major diatom-sinking event which occurs at the end of the regional bloom is driven by physiologically deteriorated cells. Here we quantify, using the Bottle-Net, Arctic diatom stocks below and above the photic zone and assess their cell health status. The communities were sampled around the Svalbard islands and encompassed pre- to post-bloom conditions. A mean of $24.2 \pm 6.7\%$ SE (standard error) of the total water column (max. 415 m) diatom standing stock was found below the photic zone, indicating significant diatom sedimentation. The fraction of living diatom cells in the photic zone averaged $59.4 \pm 6.3\%$ but showed the highest mean percentages (72.0 %) in stations supporting active blooms. In contrast, populations below the photic layer were dominated by dead cells ($20.8 \pm 4.9\%$ living cells). The percentage of diatoms' standing stock found below the photic layer was negatively related to the percentage of living diatoms in the surface, indicating that healthy populations remained in the surface layer. Shipboard manipulation experiments demonstrated that (1) dead diatom cells sank faster than living cells, and (2) diatom cell mortality increased in darkness, showing an average half-life among diatom groups of $1.025 \pm 0.075$ d. The results conform to a conceptual model where diatoms grow during the bloom until resources are depleted and supports a link between diatom cell health status (affected by multiple factors) and sedimentation fluxes in the Arctic. Healthy Arctic phytoplankton communities remained at the photic layer, whereas the physiologically compromised (e.g., dying) communities exported a large fraction of the biomass to the aphotic zone, fueling carbon sequestration to the mesopelagic and material to benthic ecosystems.

## 1 Introduction

Diatoms can support most of the Arctic primary production during the spring phytoplankton bloom (Krause et al., 2018), the key event setting the ecosystem and driving the intense carbon-uptake characteristic of the Arctic (Vaquer-Sunyer et al., 2013). However, silicic acid concentrations $[\mathrm{Si(OH)}_4]$ are characteristically low in the European sector of the Arctic, due to the inflow of Si-depleted Atlantic water (Rey, 2012). In the Svalbard island region, Krause et al. (2018) showed diatoms to be limited by $[\mathrm{Si(OH)}_4]$ at the spring bloom and suggested that silicon limitation could collapse a diatom bloom before nitrogen, when spring conditions favor diatoms instead of favoring CE4 the haptophyte *Phaeocystis*. A similar observation was made during the spring bloom in southern Greenland, whereby diatom depletion of

[Si(OH)$_4$] collapsed the bloom with $\sim 4\,\mu\text{mol}\,\text{L}^{-1}$ remaining nitrate (Krause et al., 2019).

The termination of the Arctic spring bloom is characterized by rapid sinking of diatom cells, leading to high sedimentary fluxes in the spring (Oli et al., 2002 TS3; Wassmann et al., 2006; Bauerfeind et al., 2009), precluding this production from being recycled in the upper ocean. The apparent rapid sinking of the senescent diatom bloom appears to sustain the depletion of $CO_2$ in surface waters initiated by the bloom and drives strong atmospheric $CO_2$ uptake (Bates and Mathis TS4, 2009) as average $p\text{CO}_2$ values post-bloom are typically below 300 ppm – with some values as low as 100 ppm (Takahashi et al., 2002; Holding et al., 2015).

Factors regulating diatom sedimentation have been explored for decades; however, there are few published reports on the status of diatoms' health in the Arctic during blooms and on the possible role deteriorated cell health status may play in driving sedimentary fluxes. Alou-Font et al. (2016) observed large variability in the health status of phytoplankton in the Canadian Arctic, influenced by the light and temperature conditions but not by nitrate concentration – typically thought to be the main yield-limiting nutrient. Silicon limitation has been shown to affect both autolysis (i.e., cell death) and the potential to form aggregates (which facilitate sinking) in *Coscinodiscus wailesii* cultures, whereas the latter was less pronounced under nitrogen limitation (Armbrecht et al., 2014).

Because of diatoms' obligate silicon requirement, its depletion in the water column would exclusively affect their physiology and, potentially, their biogeochemical fate. Lomas et al. (2019) recently demonstrated that polar diatoms have high elemental density (i.e., element content per unit biovolume) relative to low-latitude diatoms, and this is especially true for silicon content (consistent with results from Baines et al., 2010). Therefore, short-term changes in diatom physiology, e.g., responses to nutrient stress, may favor rapid sinking of polar diatoms much more than in temperate diatom species. While one could examine diatom cells from sediment traps, which is a standard approach used to explore diatoms' sinking fluxes, this methodology precludes accurate analysis of physiological health due to both the time required to collect cells (i.e., cell status can change) and trap fixatives (necessary to avoid "swimmers" from consuming sedimented material) lead to mortality of all cells. A new instrument, the Bottle-Net, has been applied to address this methodological gap. The Bottle-Net is a plankton net fitted inside a Rosette sampling system that can be used to collect plankton samples at depth without a prolonged deployment. This system was recently used to assess the stock and health status of microplankton in deep waters across the subtropical and tropical ocean (Agustí et al., 2015). Using the Bottle-Net at stations around the Svalbard islands, we examined diatom stocks within and below the photic layer and assessed their health status along contrasting stages of bloom development. We also conducted two exploratory experiments to

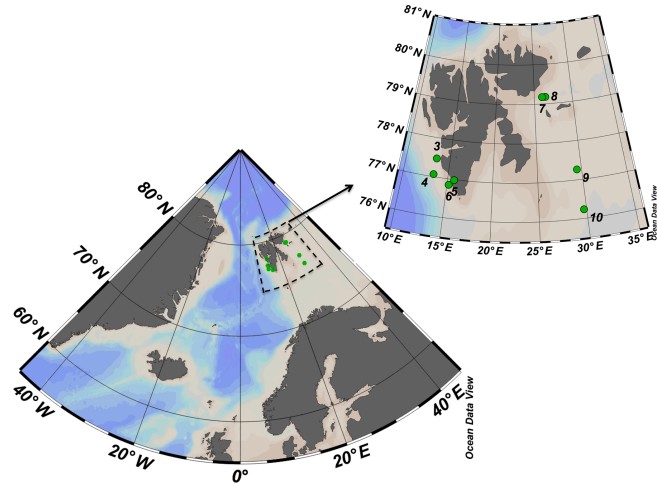

**Figure 1.** ARCEx cruise study area with the insert showing the station number and location (green dots) around the Svalbard islands. TS5

test the hypotheses that dead diatom cells in the field sink faster than living ones, based on previous culture experiment results (Smayda, 1971), and that spring field diatoms can die rapidly upon falling below the photic layer.

## 2 Methods

### 2.1 Sampling and study area

The study was conducted between 17 and 29 May 2016 onboard the R/V *Helmer Hanssen*. The cruise started in the southwestern fjords of Svalbard islands transited northward toward Erik Eriksen CE5 Strait and then south towards stations near the polar front and the Barents Sea (Fig. 1).

Vertical profiles with a Seabird Electronics 911 plus CTD, provided with an oxygen sensor, fluorometer, turbidity meter and PAR sensor (Biospherical/LI-COR, SN 1060), were conducted at all stations sampled. Water samples were collected using 12 5 L Niskin bottles installed on a rosette sampler. Water samples were taken between the surface and the bottom (max. 500 m) for analysis of nutrients, diatom silica, productivity and other properties (Krause et al., 2018). We calculated the upper mixed layer (UPM) as the shallowest depth at which water density ($\sigma_\theta$) differs from surface values by more than $0.05\,\text{kg}\,\text{m}^{-3}$ (Mura et al., 1995).

At eight of the stations (Fig. 1), microphytoplankton samples were collected by using two Bottle-Net devices installed on the rosette sampler. The Bottle-Net is a new oceanographic device developed for the Malaspina 2010 circumnavigation expedition, which consists of a 20 μm conical plankton net housed in a cylindrical PVC pipe and is designed to be mounted in place of a Niskin bottle on the rosette sampler. The Bottle-Net cover (on top) hermetically opens and closes bottle remotely using the rosette's carousel bottle-

firing mechanism, thereby initiating or terminating sample collection; CE6 the casing is open at the bottom to allow the water filtered through the internal plankton net to flow out (Agustí et al., 2015). The Bottle-Net is lowered with the top cover closed, opened at the desired bottom depth ($D_b$, m) during the ascension of the rosette, and then the top cover is closed again at the upper depth ($D_u$, m) of the water column to be sampled. This results in one integrated sample, from $D_b$ to $D_u$, per deployment. Two Bottle-Nets were used mounted in the rosette sampling system, one to collect phytoplankton at the aphotic zone and the second to collect the community in the upper water column (photic layer). The two layers were selected by combining the information on light penetration (PAR sensor) and chlorophyll *a* fluorescence obtained during the downward CTD cast. The upper layer included the thickness of the photic layer to the depth where chlorophyll fluorescence faded away, which typically corresponded to very low levels of PAR (e.g., $\sim 0.1$ %–1 % of surface irradiance). For the aphotic zone, one Bottle-Net was remotely opened and started filtering water when the rosette reached the maximum depth at each station, and it collected until reaching the depth 10 m below the maximum depth of the photic zone. The second Bottle-Net was opened at the bottom of the photic layer and was kept CE7 open until reaching the water surface. Once on deck, the Bottle-Nets were gently rinsed with filtered seawater to retrieve the sample from the collector. Sampled volume was estimated as the product between the cross-sectional area of the mouth of the Bottle-Net and the vertical distance covered by the device from the start of the ascension to the closure of the top cover ($D_b$ to $D_u$). The Bottle-Net presents an aspect ratio of 4, to avoid resuspension of materials filtered, displaying an efficiency of filtration of 96 % for deep tows (2000–4000 m) at towing velocities around to 30 m min$^{-1}$, i.e., standard rosette retrieval velocities (Agustí et al., 2015).

## 2.2 Microplankton abundance and viability

Bottle-Net subsamples were processed to identify living and dead phytoplankton cells. The freshly collected samples were filtered onto 0.8 µm pore size black Nuclepore filters, stained with the BacLight™ Viability kit, placed on slides and frozen at $-80\,°C$ until examination under epifluorescence microscopy. Another fraction of the sample collected by the Bottle-Net was fixed with formalin for further examination at the laboratory. The observed diatoms were classified to genera. The percentage of living or dead cells relative to the total (i.e., dead plus living) was calculated for the total community and by genera.

The BacLight™ Viability kit (Molecular Probes™, Invitrogen) is a double staining technique to test cell membrane permeability and is proven to be an effective method for determining phytoplankton viability (Llabrès and Agustí, 2008; Agustí et al., 2015). When excited with blue light under the epifluorescence microscope, living phytoplankton cells with intact membranes fluoresce green (SYTO 9, nucleic acid stain) and dead phytoplankton cells with compromised membranes fluoresce red (propidium iodine, nucleic acid stain). The samples were examined under blue light, most onboard the research vessel, using a Partec CyScope® high-power blue (470 nm) and green (528 nm) LED-illuminated epifluorescence microscope. In the laboratory at KAUST, all samples were examined using a Zeiss Axio Observer Z1 epifluorescence LED-illuminated microscope (Colibri 7 LED system). The fluorescence of the stained cells is well preserved at $-80\,°C$ for several months, and samples were transported frozen between the port of arrival (Tromsø, Norway) and KAUST.

## 2.3 Decay and sinking rates of living microphytoplankton cells

The expected mortality rates of living phytoplankton cells when transferred from the photic to the aphotic layer were examined at station no. 3 with vertical tows from the photic layer. An aliquot of the photic-layer microphytoplankton sample was resuspended in 2 L of 0.7 µm filtered surface water and incubated in the dark at $4\,°C$ for 7 d, simulating sedimentation from the photic layer into the aphotic layer. The community was sampled at the onset of the experiment and during set time intervals (i.e., 1, 3, 5 and 7 d). Immediately after sampling, cells were stained with the vital stain BacLight™ kit, then prepared and examined under an epifluorescence microscope (as described above) to quantify the proportion of living cells in the community. The half-life (i.e., the time required for the number of living cells to decline by 50 %) and the decay rate for each living-cell population were then calculated from the decline in living cells over time.

An experiment to test whether dead diatom cells sink faster than living cells was conducted shipboard using a sinking column (30 cm diameter, 1.35 m height, internal volume of 95 L). The chamber was placed on deck, filled with 20 µm filtered surface seawater, and left undisturbed for $\sim 1$ h before starting the experiment. Microplankton collected in a vertical net tow (20 µm mesh) from the photic layer of Erik Eriksen Strait (close to the position of station no. 7) was resuspended in 1 L of 0.7 µm filtered seawater and gently added at the surface of the sinking column. A fresh subsample of the initial community, which was added to the surface of the chamber, was stained with the BacLight™ kit and the diatoms were examined for identification and the percentage of living or dead cells as described above. The samples at the bottom of the sinking column (sampling port located 1.35 m below the surface) were collected at intervals of time of 0 (time when the sample was added at the surface), 1, 4 and 12 h after the initial time, and were processed similarly to the initial community material.

## 3   Results

The stations sampled encompassed a spectrum of bloom conditions. Station 4 (off the western Svalbard shelf) waters were pre-bloom, as indicated by low diatom stocks, high dissolved inorganic nutrient concentrations (photic layer concentrations $Si(OH)_4 = 4.15 \pm 0.04\,\mu mol\,Si\,L^{-1}$, $NO_3 + NO_2 = 9.43 \pm 0.09\,\mu mol\,N\,L^{-1}$, Table 1) and relatively low stratification (Table 1). All other stations sampled were characterized by comparatively depleted nutrient concentrations (photic layer concentrations $Si(OH)_4 = 0.99 \pm 0.30\,\mu mol\,Si\,L^{-1}$, $NO_3 + NO_2 = 1.93 \pm 0.76\,\mu mol\,N\,L^{-1}$, Table 1), thereby representing communities that were either in advanced blooming stages or were senescent after blooming. Stations 6 (SW Svalbard shelf) and 8 (E Svalbard shelf) supported actively blooming diatom populations, with station 8 having the highest chlorophyll $a$ concentration ($10.5\,\mu g\,Chl\,a\,L^{-1}$ CE8, as described in Krause et al., 2018), and a large fraction of living diatom cells (about 70 %, Table 1). Both locations had the highest stratification among the stations, as indicated by the low UPM values (Table 1). In contrast, station 9 (polar front) supported a senescent diatom population in post-bloom phase, as indicated by depleted nutrient pools and a low percentage of living diatom cells (46.0 %, Table 1). The highest mixing was observed at station sampled at the Barents Sea (Table 1). The percentage of living cells was not significantly correlated with the concentrations of $NO_3 + NO_2$ (two-tailed test, $r = -0.54$, $P = 0.17$) or $Si(OH)_4$ (two-tailed test, $r = -0.69$, $P = 0.06$).

Taxonomic classification under epifluorescence microscopy is not particularly accurate, but we were able to unambiguously identify different diatom genera and some species. The more abundant genera found in the samples were *Thalassiosira* CE10 spp., differentiated between large (L TS9 *Thalassiosira*) and small (*Thalassiosira*) morphotypes; *Chaetoceros* spp., with a large representation of *Chaetoceros socialis*; pennate diatoms including colonies of *Fragilariopsis* spp., *Navicula pelagica* and *Pseudo-nitzschia* spp.; less abundant but identifiable cells of *Amphiprora hyperborean*; and *Coscinodiscus* sp. among others.

The living (green fluorescence) and dead (red fluorescence) cells were clearly identified under the LED-illumination of the epifluorescence microscopes used (Fig. 2). The fraction of living diatom cells in the photic layer averaged $59.4 \pm 6.3$ % but ranged broadly from 20.9 % in station 4, in pre-bloom state, to 72.0 % in station 5, which supported an active bloom. In contrast, the population sinking below the photic layer was comprised mostly of dead cells ($20.8 \pm 4.9$ % living cells, Fig. 2). Indeed, the fraction of living diatoms was consistently greater in the photic layer than in the diatom stock sinking below the photic layer (Wilcoxon ranked sign test, $P = 0.0078$, Fig. 3), a pattern consistent across taxa found in at least four of the stations (large-celled *Thalassiosira* spp., $P = 0.02$, $N = 4$; *Fragilariopsis* spp., $P = 0.005$, $N = 6$; *Chaetoceros* spp., $P = 0.0054$, $N = 6$;

**Table 1.** Stations number and location, averaged ($\pm$ SE) photic layer temperature, salinity, upper mixed layer (UPM) depth, nutrients, and measurements made with the Bottle Net (BN) in the photic and aphotic zones, indicating the depth of the tows, and the abundance and percentage of living diatoms found at the two layers.

| Station | Latitude N TS7 | Longitude E | Temperature (°C) | Salinity (psu) | UPM (m) | $NO_3 + NO_2$ (µM) | $PO_4$ (µM) | $Si(OH)_4$ (µM) | BN photic (range, m) | BN aphotic (range, m) | Photic diatoms (cells m$^{-2}$) | Aphotic diatoms (cells m$^{-2}$) | Photic living diatoms (%) | Aphotic living diatoms (%) |
|---|---|---|---|---|---|---|---|---|---|---|---|---|---|---|
| No. 3, Bellsund Hula | 77°28.09′ | 13°27.483′ | 0.81±0.33 | 34.48±0.083 | 14.5 | 1.79±1.52 | 0.27±0.11 | 0.75±0.45 | 45–0 | 197–55 | $3.160 \times 10^7$ | $6.843 \times 10^6$ | 63.54 | 21.70 |
| No. 4, Bredjupet | 77°03.356′ | 13°23.369′ | 4.64±0.025 | 35.0±0.001 | 65.5 | 9.44±0.097 | 0.63±0.019 | 4.16±0.046 | 60–0 | 415–100 | $3.04 \times 10^5$ | $1.63 \times 10^5$ | 20.93 | 9.47 |
| No. 5, Inngang Hornsund | 76°58.73′ | 15°44.113′ | −0.54±0.035 | 34.27±0.037 | 24.2 | 5.66±0.019 | 0.34±0.078 | 2.45±0.40 | 50–0 | 220–80 | $3.20 \times 10^7$ | $3.00 \times 10^5$ | 72.03 | 0.50 |
| No. 6, Hornsund Dypet | 76°51.244′ | 15°13.143′ | −0.034±0.1 | 28.98±4.5 | 9.8 | 0.49±0.37 | 0.17±0.03 | 0.36±0.118 | 50–0 | 220–60 | $2.01 \times 10^9$ | $4.69 \times 10^8$ | 70.03 | 8.31 |
| No. 7, Erik Eriksen Strait CE9 | 79°09.986′ | 26°02.20′ | −1.44±0.093 | 34.29±0.04 | 35.9 | 0.03±0.026 | 0.16±0.01 | 0.07±0.012 | 50–0 | 260–70 | $1.25 \times 10^7$ TS9 | $1.13 \times 10^6$ | 61.12 | 26.99 |
| No. 8, Erik Eriksen Strait | 79°10.479′ | 26°27.518′ | −1.31±0.088 | 34.22±0.4 | 3.0 | 2.23±1.64 | 0.15±0.077 | 0.57±0.40 | 50–0 | 245–70 | $2.47 \times 10^{10}$ | $5.56 \times 10^6$ | 69.79 | 31.27 |
| No. 9, Polar front | 77°15.308′ | 29°29.243′ | 2.04±0.099 | 34.7±0.027 | 34.0 | 0.14±0.034 | 0.204±0.022 | 1.29±0.17 | 50–0 | 180–60 | $2.76 \times 10^7$ | $2.27 \times 10^6$ | 45.97 | 50.00 |
| No. 10, Barents Sea | 76°13.513′ | 29°43.710′ | 4.06±0.044 | 34.9±0.001 | 75.0 | 3.21±0.20 | 0.345±0.03 | 1.48±0.156 | 50–0 | 180–60 | $1.45 \times 10^8$ | $2.35 \times 10^7$ | 71.77 | 13.14 |

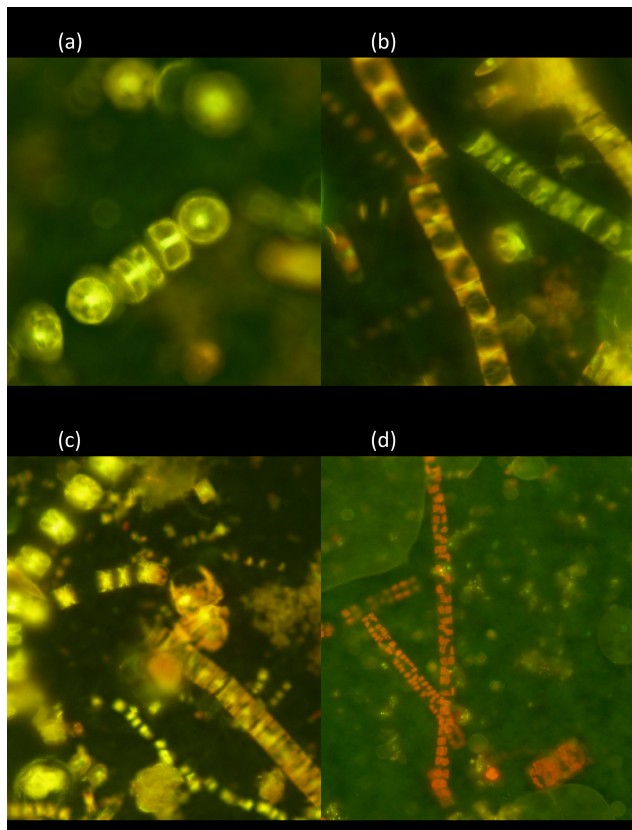

**Figure 2.** Photographs of the natural Arctic diatoms sampled with the Bottle-Net observed under epifluorescence microscopy and stained with the BacLight™ kit. **(a)** Colonies of *Thalassiosira* spp. showing green fluorescence corresponding to living cells. **(b)** Colonies of *Fragilariopsis* spp. showing dead cells (red fluorescence, vertical-left colonies) and living cells (green fluorescence, transversal-right colony). **(c)** Surface layer community, composed of multiple diatom genera (*Chaetoceros* spp., *Fragilariopsis* spp., *Thalassiosira* spp., pennates), showing a blend of living cells (green fluorescence) and dead cells (red fluorescence). **(d)** Aphotic zone sample showing dead colonies (red fluorescence) of *Fragilariopsis* spp. and *Thalassiosira* spp. (two-cell colony in the bottom right of the photo).

Fig. 3), but the percent living cells in the photic layer and below this layer was not significantly different for the small-celled *Thalassiosira* spp. ($P = 0.09$, $N = 6$).

Among stations, there was significant variability in the diatom assemblage structure. Earlier cruise stations were dominated by *Fragilariopsis* spp. and *Chaetoceros* spp. This changed from station 6 to 8, where communities were dominated by *Fragilariopsis* spp. and *Thalassiosira* spp. and were the areas with the highest diatom biomass observed (station no. 8, Fig. 4). Community composition changed at the polar front and Barents Sea stations (Fig. 4) with a larger contribution of other taxa, including *Navicula pelagica* (station no. 9, Fig. 4). The diversity of the diatoms found at the aphotic zone differed in several stations from that found at the photic layer (Fig. 4). The large-celled *Thalassiosira* sp. colonies dominated the aphotic community in several stations although they were not dominant at the photic community (Fig. 4). At station no. 4, the community sampled was more diverse at the aphotic layer than at the photic layer (Fig. 4). The stock of diatoms that had sunk below the photic layer comprised, on average, $24.2 \pm 6.7\%$ of the total water column stock, with the proportional contribution ranging considerably among groups (Fig. 5). The proportion of biomass of the large-celled *Thalassiosira* colonies in the aphotic layer was the largest and *Chaetoceros* spp. the smallest (Fig. 5). Station no. 4 (pre-bloom status) had a larger proportion of diatom biomass in the aphotic layer, and station no. 8, diatom bloom station, had the lowest. At station no. 8, however, the photic-zone population of the dominant *Thalassiosira* species contained 54.8% of living cells and was paralleled with a significant contribution of dead cells at the aphotic layer (Fig. 4), suggesting the collapse of the bloom had already been initiated despite the considerable photic-layer biomass. Similarly, *Fragilariopsis* senescence at the photic layer of station no. 3 (only 35.1% of cells were alive at the photic layer) helps explain its larger contribution than in the aphotic layer (Fig. 4). There was a significant negative relationship between the percent of the diatom stock population that had sunk below the photic layer and the percent of living cells in the photic layer ($R^2 = 0.39$, $P < 0.001$, Fig. 5b), indicating that healthy, actively growing populations largely remain in the surface.

The suggestion that dead diatom cells sink faster than living cells was tested experimentally. Initially, only 6.7% of the cells of the *Flagilariosis* spp. and *Thalassiosira* spp. colonies dominating the community tested were dead. However, all cells settling to the bottom of the sedimentation chamber within 1 h of the experiment start were dead, including large *Coscinodiscus* sp. cells (Fig. 6). The population of cells settling to the chamber bottom 4 and 12 h following addition of the fresh, healthy community, was also largely dominated by dead *Flagilariosis* spp. and *Thalassiosira* spp. colonies, 82.2% and 71.7%, respectively. And the fraction of living cells which had settled the height of the chamber proportionally increased with time (Fig. 6). These experimental results indicated that dead diatom cells among the groups examined sink faster than living cells.

The experiment testing diatom survival in aphotic zone indicated that once diatom cells sink below the photic layer, they would die rapidly. The incubation was performed close to the temperature below the photic layer that averaged $2.978\,°C \pm 0.003$ at station no. 3, suggesting no thermal effects. The cell concentration at the onset of the experiment was $298\,\text{cells}\,\text{mL}^{-1}$. The half-life (i.e., percent of living cells reduced to half) survival times were remarkably uniform across diatom taxa, ranging from 0.9 d, for *Thalassiosira* spp. to 1.3 d for *Coscinodiscus* sp., depending on species (Fig. 7). Once dead, the cells lysed; half-life periods for cell death

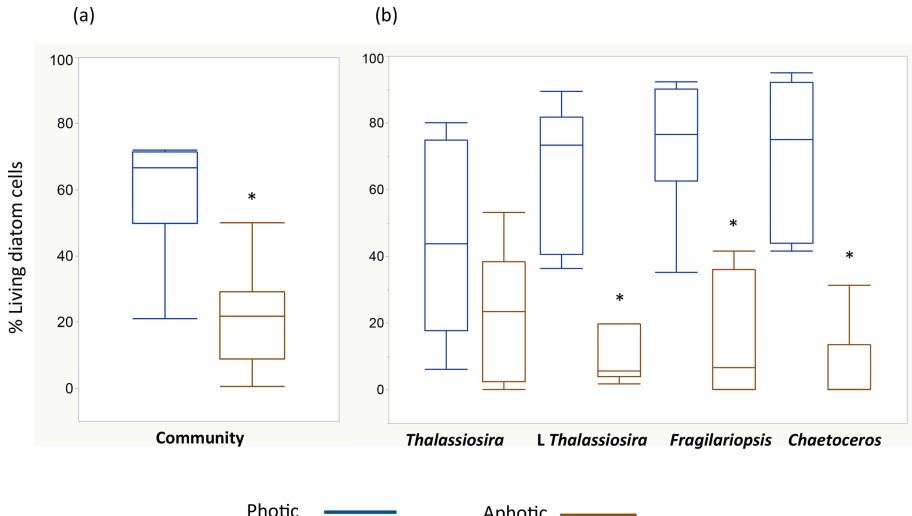

**Figure 3.** Box plots showing the distribution of the percentage of living diatoms in the photic CE11 (blue) and aphotic (brown) layers. Percentage of living cells among **(a)** the total diatom community and **(b)** for the populations of the most abundant diatom taxa observed during the cruise. The asterisks indicate significant differences between photic and aphotic zones ($P < 0.05$). Boxes encompass the central 50 % of the data, the horizontal line inside the box represents the median, and vertical bars encompass 90 % of the data.

and lysis after transfer into aphotic darkness increased from 1.6 d, for the smaller *Flagilariosis* spp. cells, to 5.3 d for the larger CE13 *Thalassiosira* spp. cells (Fig. 7).

## 4   Discussion

The results presented confirm that active and healthy diatom populations, as those actively growing during the spring bloom, are associated with relatively small stocks of fast-sinking diatoms. In contrast, unhealthy diatom populations, such as those present before blooming has initiated or in the senescent phase of the bloom, characterized by a large fraction of dead cells, support comparatively larger pools of sinking diatoms.

These observations are consistent with early reports, based mostly on laboratory cultures, indicating that dead diatom cells sink faster than living ones (Smayda, 1971). The experiment conducted, albeit at only one station, showed that dead cells sank much faster than living ones in a field assemblage with considerable diversity in species and in the physiological condition. Indeed, whereas the dominant populations tested were dominated by living healthy cells, only dead cells were collected at the bottom of the sedimentation chamber over the first few hours of the experiment, and the proportion of living cells collected increased over time. Moreover, our experimental assessment of diatom survival incubated at aphotic conditions suggested that once sinking below the photic layer, diatom cells could die at half-lives of 21.8 to 30.2 h across species. This result, although limited to one experiment, was consistent among the major genera and functional groups analyzed, and it reflected sur-

vival at in situ conditions. Smayda and Mitchell-Innes (1974) also reported the decrease in viable cells after darkness: "After 6 days of dark incubation, the number of viable cells of *Chaetoceros curvisetus* recognizable decreased from 760 to 240 cells per ml.", representing a decay rate of $0.19\,\mathrm{d}^{-1}$ (i.e., 50 % loss of cells in 3.6 d) comparable to the rate reported here. Other studies also reported rates of living cell mortality in darkness close to those found here (Segovia and Berges, 2009; Agustí et al., 2015). The decay rates calculated for living or viable vegetative cells were more than 3 times faster than those observed for the total cells in the population. These included both viable and non-viable vegetative cells, which are, however, morphologically similar and could not be differentiated unless using specific methods to discriminate living from dead cells, such as the staining test used here. Phytoplankton vegetative cells do not survive under darkness (Smayda and Mitchell-Innes, 1974; Segovia and Berges, 2009; Segovia et al., 2003), and only resting spores and resting cells are able to survive in the dark (Ignitiades TS10 and Smayda, 1970; Smayda and Mitchell-Innes, 1974; Peters and Thomas, 1996). Our results reporting fast diatom cell death under aphotic conditions are contrasting with the expectation of high survival capacity of polar diatoms to darkness supported by existing evidence. Recent reports identified fast photosynthetic response to irradiance in diatoms sampled during the dark wintertime around the Svalbard islands (Kvernvik et al., 2018 TS11). Phenotypic selection for specific physiological properties allows polar diatoms to acclimate to low light and darkness (Lacour et al., 2019). Our experiment, however, was carried out CE15 in a spring community under a 24 h light : 0 h darkness photoperiod and with shallow mixing depths. The community tested

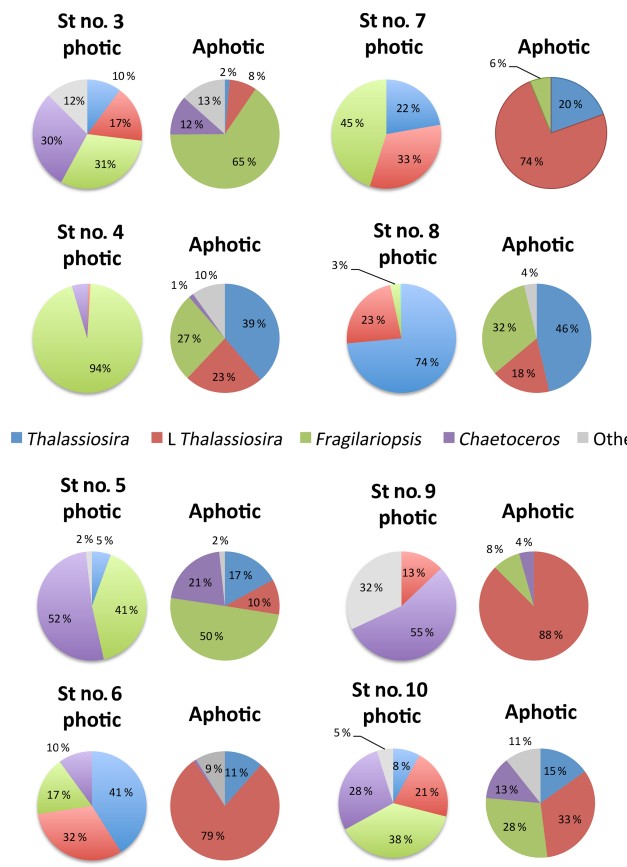

**Figure 4.** Pie charts showing the diatom community within the photic and aphotic zones among stations. Colors correspond to different taxa, and the numbers indicate the percentage of cells relative to the total community.

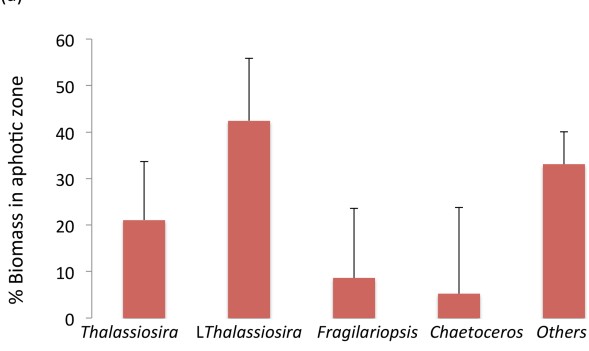

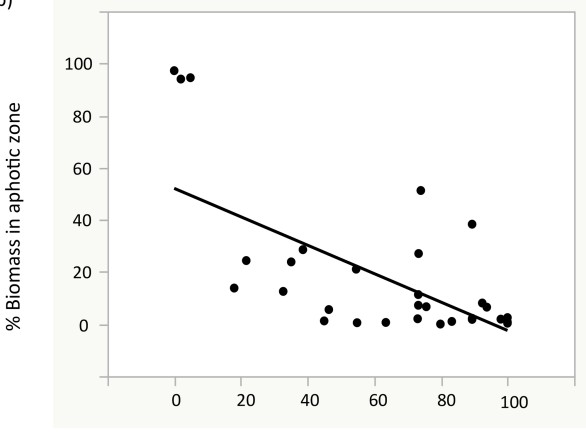

**Figure 5.** **(a)** The proportion (mean $\pm$ SE) of the water-column stock observed in the aphotic zone (relative to the sum of the aphotic and photic zones) among the different diatom taxa. **(b)** The relationship between the percentage of living diatoms cells among the different populations in the photic layer and the proportion of the water-column population stock observed in the aphotic zone. The line represents the fitted linear regression ($R^2 = 0.39$, $P < 0.001$).

was, therefore, likely photo-acclimated to a prolonged photoperiod and relatively high irradiance. Therefore, it is expected to respond differently under darkness than Arctic microalgae growing underneath the ice or under very short photoperiods and/or minimal irradiance levels (Lacour et al., 2019; Berge et al., 2015). Hence, Arctic phytoplankton are expected to show contrasting responses to prolonged darkness in the spring, when acclimated to longer photoperiods than in winter in which surviving cells are expected to be acclimated to short photoperiods.

Beyond the stressor of continuous darkness, the fast decay rates observed here under aphotic experimental conditions could also have been influenced, in addition to darkness, by the low nutrient availability and/or the possible presence of pathogens or parasites. This suggests CE16 that survival of natural populations below the photic layer may be lower than expected in the dark from axenic, high-nutrients in vitro studies with cultures, an aspect already pointed by Ignitiades and Smayda (1970). The [Si(OH)$_4$] at station no. 3 was $0.75\,\mu\mathrm{mol}\,\mathrm{Si}\,\mathrm{L}^{-1}$ and [NO$_3$ + NO$_2$] was $1.79\,\mu\mathrm{mol}\,\mathrm{N}\,\mathrm{L}^{-1}$; Lomas et al. (2019) demonstrated that po-

lar diatoms have higher Si : N requirements ($> 1.5\,\mathrm{mol} : \mathrm{mol}$) than lower-latitude diatoms; thus nutrient data CE17 would predict Si would be the yield limiting relative to N. As the decline rates were derived from a single test, further experiments on cell decay rates of Arctic diatoms sampled in the spring under dark conditions will be required to confirm our results.

Cell abundance and health state observed were consistent with previous studies. In the Canadian Arctic, living cells in open water and ice-covered stations represented $57.3 \pm 5.8\,\%$ and $48.0 \pm 3.9\,\%$ ($\pm$ SE), respectively (Alou-Font et al., 2016), which are CE18 similar to the proportions in our study. The percentage of living cells was higher during the bloom periods than the periods before and after (Alou-Font et al., 2016). This result appeared to be driven by light and low nutrient concentrations (Alou-Font et al., 2016). The quantification of the percent of living cells in our study helped to identify the different stages of the Arctic spring bloom

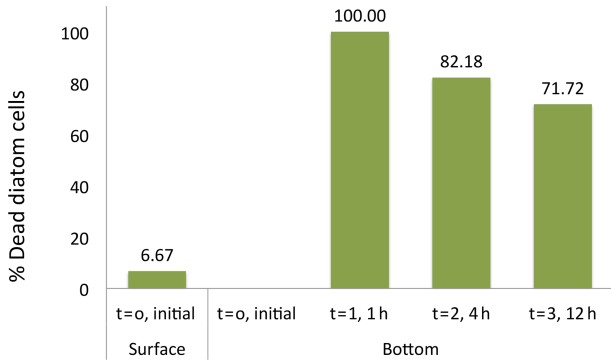

**Figure 6.** Diatom cell viability among the sinking cells. The initial percentage of dead cells correspond to the fresh photic-zone Arctic microplankton sample (see Methods) and added to the surface of the sinking column (1.35 m height) at time 0. The percentages of dead cells at the bottom of the sinking column were collected at intervals times of 0, 1, 4 and 12 h after sample inoculum addition.

among the stations sampled. A pre-bloom situation with low cell abundance and a small percentage of living cells was found at station no. 4 located further west of Svalbard islands, where silicic acid and nitrogen concentrations were higher and mixing was more significant than in other Arctic stations. The healthiest diatom community was observed at station no. 5, where the high stratification and $Si(OH)_4$ concentration above the half saturation constant ($K_S$TS12) of $2 \mu M$ (from kinetic experiments in the same region by Krause et al., 2018) helped the diatoms to grow actively. The highest cell abundance was observed at station no. 8, but this persisted with a lower percentage of living diatoms, and the $Si(OH)_4$ concentration well below the $K_S$ value indicated that the bloom was reaching the maximum capacity, although diatom sinking was still low. A post-bloom situation was identified at the polar front community, with similar percentages of living cells at the photic and aphotic zones as a result of high sinking induced by Si and nitrogen limitation, as suggested by the lower $Si(OH)_4$ $K_S$ of $0.8 \mu M$ (Krause et al., 2018). The diatom community captured by the Bottle Net below the photic layer was consistent with the limited, but comparable, data obtained by sediment traps deployed in the area which also indicated *Fragilariopsis* and *Thalassiosira* species to be the dominant contributors to Si and biomass export (Krause et al., 2018).

Given the range of bloom development represented among our stations, the results presented conform to a conceptual model in which nutrients, including Si (Rey, 2012; Krause et al., 2018), and a mixed layer drive the growth of diatoms during the Arctic spring bloom (Wassmann et al., 1997; Reigstad et al., 2002). For diatoms, Si depletion results in two potential physiological issues: yield limitation (i.e., diatom standing stock is too high to be supported by the available silicic acid) and intense kinetic/ CE19 growth limitation (i.e., depleted silicic acid limits diatom Si uptake to such a degree that growth

must slow, Krause et al., 2018). Thus, such a situation would stimulate mass sedimentation, suggested to be an evolutionary adaptation to help diatom communities persist when nutrients are limiting (Raven and Waite, 2004). A large fraction (30 %–50 %) of the diatom cells in the silicon-depleted photic layer were dead, pointing at acute silicic acid limitation as the likely factor triggering partial mortality, while the remaining cells are likely to be senescent. Unhealthy diatoms would then lose the capacity to actively regulate buoyancy that characterizes healthy diatom cells (Smayda, 1970), leading to rapid sinking of the bloom. Acute silicic acid limitation is identified, therefore, as the event leading to loss of the capacity to actively regulate buoyancy that characterizes diatom cells (Smayda, 1970) and rapid sinking of the bloom. The potential for rapid sedimentation is enhanced by higher silica quotas for polar diatoms (Lomas et al., 2019) compared to lower-latitude diatoms (Brzezinski, 1985).

Diatoms have been shown to have a remarkable metabolic capacity to regulate buoyancy (Gemmel et al., 2016 TS13), both maintaining zero (Gemmel et al., 2016) and positive buoyancy (e.g., Villareal et al., 2014) involving regulation through the production of osmolytes (Gradmann and Boyd, 2002), which plays an important role in exploiting nutrient patchiness within the photic layer (Villareal et al., 2014). Diatom sinking rates are inversely related to growth rate (Gemmel et al., 2016), so that silicon depletion is expected to result in increased sinking rates, despite field diatoms reducing their silica per cell when kinetically limited by silicic acid (McNair et al., 2018). There is experimental demonstration that silicon depletion plays the most important role, compared to nitrogen or phosphorus, in triggering rapid sinking of diatom cells, indicating that biochemical aspects of silicon metabolism are particularly important to diatom buoyancy regulation (Bienfang et al., 1982). $N : P$ ratios in this region do not suggest that phosphorus plays a limiting role in primary production, and when silicic acid is depleted enough nitrate remains to fuel growth of other phytoplankton groups (e.g., *Phaeocystis*, Krause et al., 2018). Once diatoms lose their capacity to regulate buoyancy and sink below the photic layer, they die rapidly and are unable to ascend back to the photic layer, resulting in the rapid sinking fluxes that drives high sedimentation rates characteristic of the termination of the Arctic spring bloom (Oli et al., 2002; Wassmann et al., 2006; Bauerfeind et al., 2009). Rapid sinking of the Arctic spring bloom, in turn, precludes carbon recycling in the photic layer, thereby leading to undersaturated $pCO_2$ driving the large atmospheric $CO_2$-uptake characteristic of the European sector of the Arctic during this season which does not equilibrate with the atmosphere until months later (Bates et al., 2009; Takahashi et al., 2002; Holding et al., 2015).

A large fraction of the total water column phytoplankton biomass was observed below the photic layer, representing on average $24 \% \pm 6.7$ ($\pm$ SE) of the surface diatom populations in the study area. This considerable proportion can be explained by high diatom export from the photic zone,

**Biogeosciences, 16, 1–11, 2019**                                                                    **www.biogeosciences.net/16/1/2019/**

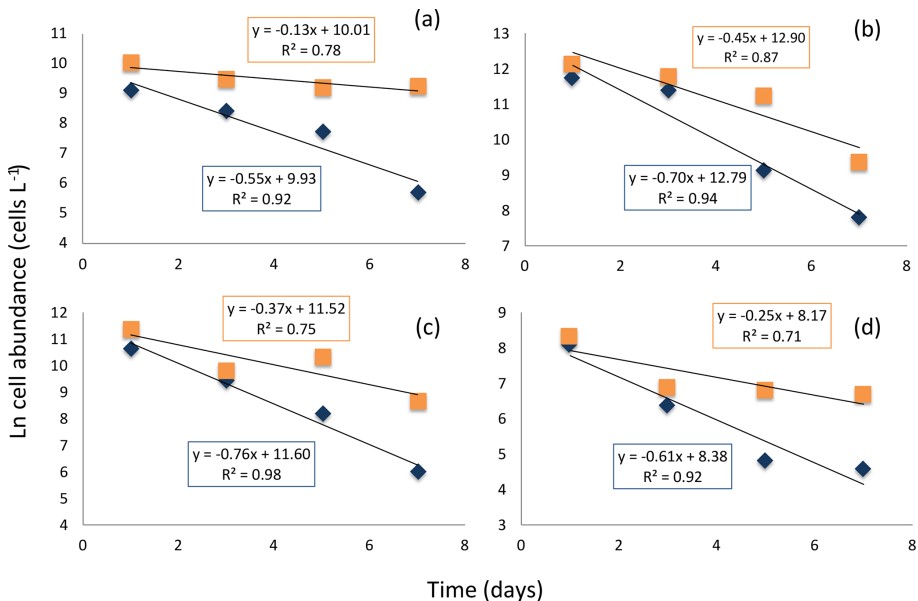

**Figure 7.** Decay in the cell abundance of living (blue diamonds) and total (orange squares) diatoms when exposed to aphotic zone light conditions. **(a)** Large-celled *Thalassiosira* spp. CE14 sp. **(b)** *Fragilariopsis* spp. **(c)** *Thalassiosira* spp. **(d)** Pennate diatom. The solid black lines and equations show the fitted linear regressions for the percent of living cells (blue box, all fitted lines significant, $p < 0.05$) and total population cells (orange box, none of the fitted lines were significant, $p > 0.05$).

as opposed to lateral advection. This is consistent with the high rates of biogenic silica (proxy for diatom biomass) export at stations 4, 7–8 and 10; rates were a factor of 4 higher than integrated diatom silica production in the upper water column and represented up to 40 % of the integrated diatom silica standing stock (Krause et al., 2018). These cruise trends are in agreement with the observation of large sinking events in the Arctic as reported for ice diatoms (Boetius et al., 2013 TS14; Aumack et al., 2014) associated with ice melting in the Arctic, which represent a large carbon supply to benthic communities in the Arctic shelves (Moran et al., 2005; Tamelander et al., 2006). While we do not report data for an ice-diatom assemblage, data for the same cruise showed that CE20 silicon-uptake rates of ice diatoms near stations nos. 7 and 8 were strongly limited by $Si(OH)_4$ concentration in the surface waters, likely limiting their growth to a degree CE21 (Krause et al., 2018), and previous studies have noted their susceptibility to silicon limitation (Cota et al., 1990; Smith et al., 1990). Our results show that healthy phytoplankton communities remained at the photic layer, although dying communities exported a large fraction of the biomass (up to 65 %) to the aphotic zone.

In summary, the results presented here support a link between diatom cell health status, likely driven by progressive nutrient limitation, and sedimentation fluxes in the Arctic. Whereas the link between diatom health status and sinking rates has long been established (Smayda, 1971), the evidence corresponded to algal cultures in the laboratory. This is the first demonstration of these ideas using natural diatom communities in this region, partially due to the logistical challenges of assessing both viability and settling in the field. Our conceptual model suggests that deterioration of diatom health, such as that occurring when acute silicon limitation CE22 or other resource limitations along the spring bloom are reached, leads to loss of the capacity to regulate buoyancy and leads to rapid sinking, with cells exported below the photic layer. Understanding the role of cell health status, and the role of silicic acid depletion, in the regulation of diatom sinking rates is fundamental to mechanistically understand the biological pump in the Arctic and its response to future changes.

*Data availability.* Data are available upon request to the authors or are available through the UiT research data bank (https://dataverse.no/dataverse/uit, last access: TS15).

*Author contributions.* SA and CMD conceived and designed the study CE23. SA, CMD, IAM, JWK, PW and SK conducted analysis. All co-authors contributed to the writing of the paper, led by SA.

*Competing interests.* The authors declare that they have no conflict of interest.

*Acknowledgements.* This research was supported by King Abdullah University of Science and Technology through baseline funding BAS/1/1072-01-01 and BAS/1/1071-01-01 to Susana Agustí and Carlos M. Duarte, respectively. CE24 The ARCEx project funded by industry partners and the Research Council of Norway (project no. 228107) provided funding to Paul Wassmann, and the Dauphin Island Sea Lab funded Jeffrey W. Krause. We thank the science team and crew of the R/V *Helmer Hanssen*, as well as S. Øygarden TS16, E. Kube, A. Renner, D. Vogedes, H. Foshaug, S. Acton, D. Wiik, B. Vaaja and W. Dobbins for logistic support.

*Financial support.* This research has been supported by the KAUST (grant nos. BAS/1/1072-01-01 and BAS/1/1071-01-01), CE25 the Research Council of Norway (grant no. 228107) and the Dauphin Island Sea Lab (internal funding). TS17

*Review statement.* This paper was edited by Koji Suzuki and reviewed by three anonymous referees.

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

## Remarks from the language copy-editor

CE1   Please note the slight edit.
CE2   Please note the slight edits.
CE3   Please note that this manuscript has undergone copy-editing according to the standards of American English.
CE4   Please check that the meaning of your sentence is intact.
CE5   Please see comment CE9.
CE6   Please check that the meaning of your sentence is intact.
CE7   Please confirm the change.
CE8   Please check that the meaning of your sentence is intact.
CE9   To be consistent with the other Norwegian names of the station sites, would you like to refer to this as Erik Eriksenstretet?
CE10   Please check that spp. and sp. are used correctly throughout the entire paper.
CE11   Please confirm the change.
CE12   Should this be *Thalassiosira* spp.?
CE13   Please check that the meaning of your sentence is intact.
CE14   Is spp. sp. correct here?
CE15   Please confirm if my addition to your sentence is ok.
CE16   I had to make this very long sentence into two shorter sentences.
CE17   Please reword this sentence as it does not entirely make sense, in particular the following part: "would predict Si would be the yield limiting relative to N".
CE18   Please check that the meaning of your sentence is intact.
CE19   Do you mean here "kinetic or growth"?
CE20   Please confirm the change.
CE21   Please confirm the change.
CE22   Please check that the meaning of your sentence is intact.
CE23   I removed "conducted the analysis" as it is written again in the next line.
CE24   Please note the edits to this section.
CE25   Please note the slight edits to this section.

## Remarks from the typesetter

TS1   If possible, please provide department.
TS2   Please confirm.
TS3   Should it be "Oli" or "Olli" throughout the paper?
TS4   Please confirm.
TS5   The composition of Figs. 1, 4, and 6–7 has been adjusted to our standards. This also includes language adjustments to Figs. 4, and 6–7. If the size of any of the figures should be increased, please let me know.
TS6   Please provide a suitable running title.
TS7   Please confirm addition of symbols.
TS8   Please confirm changes.
TS9   Should the L also be italic?
TS10   Should it be "Ignitiades" or "Ignatiades" throughout the paper?
TS11   Changed 2019 to 2018; please confirm.
TS12   Please confirm.
TS13   Not mentioned in the reference list. Please check throughout.
TS14   Changed 2014 to 2013; please confirm.
TS15   Please provide date of last access.
TS16   Please provide full first names throughout the acknowledgements.
TS17   Please note that the funding information has been added to this paper. Please check if it is correct. Please also double-check your acknowledgements to see whether repeated information can be removed or changed accordingly. Thanks.
TS18   Please provide page range or DOI and article number.
TS19   Please provide all author names.
TS20   Please check the number.

TS21  Please check the number.
TS22  Please check the number.
TS23  Please check DOI and provide volume and page range.
TS24  Please provide all author names.
TS25  Please check the number.
TS26  Please provide all author names.
TS27  Please provide DOI.