# Peer review of "Arctic (Svalbard Islands) Active and Exported Diatom Stocks and Cell Health Status"

_Biogeosciences, 2018_

## Referee Comment (RC1) · Anonymous Referee #1 · 11 Dec 2018

The manuscript describes the health status of diatoms during the course of the bloom in the Arctic. The main finding is that when diatoms are dying they sink out of the photic zone. Two main types of results are described here. First a clear and complete description of the diatoms in 8 stations around svalbard, and second an experiment testing the decay of diatoms in the dark, while comparing the sinking of living versus dead diatoms. While I feel that these data are very intersting, the findings are not new and should have been presneted with others in order to give a valuable manuscript. The discussion is a little weak and rely a lot on the paper by Krause et al. For example, the discussion starts saying that diatoms in Arctic are limited by silicates and that silicates depletion is the driver of diatom death and sinking which is a result from the study by Krause et al. 2018. Why didn't you use the results of this study regarding the survival

of diatoms in the dark? Can't it be one of the trigger if the mixing increase? The paper states that the average life of the diatoms in the dark is slightly superior than a day. In this part of Arctic I guess that there is strong mixing. How long are the diatoms kept in darkness due to mixing? The data from station 9 (polar front) showed indeed that there is an effective mixing (similar diatom concentrations and % of living cells in photioc and aphotic samples), however, the % of living cells is still high. How do the authors explain that?

The different stations are ideally located and sampled to describe the diatom bloom from the initiation to the decline, but these could be more interestingly discussed in the paper. What can be brought to light from the results of this paper?

What is the bloom status at each station at the sampling time? this could be a lot more discuss using diatom cell concentrations in photic and aphotic zone, % of living cells, nutrients concentrations.... How are the nutrient concentrations compared to the winter concentrations ? that may give an idea of the bloom advancement. How is the bloom terminate? Why these data are not in the paper by Krause et al if it uses so much of the conclusions isuued from it? Alone I feel that these data even if very interseting are too poor. What are the limitations there? Why do the authors state that there is only silicate limitations and not nitrate while nitrate are also very depleted in some zone (station 6, 7 and 8) It would have been great to discuss them in light with production rates, limitations or sinking fluxes of bSi or POC from sediment traps data.

---

## Referee Comment (RC2) · Anonymous Referee #2 · 17 Dec 2018

This MS sheds a light on the role and fate of diatoms over a course of a spring bloom in the Arctic Ocean, based on the estimates of their mortality, senescent rate, and the population with fast sinking rate. These estimations were designed to test a hypothesis in which Si-depletion triggers (1) senescence of diatoms and (2) selective sinking of the dying population. Because of intense $CO_2$-sequation in the Arctic Ocean, this hypothesis is valuable to be tested, but the results in this study unlikely support this hypothesis. For example, high % living diatoms in the upper layer was achieved at Stns 6, 7 and 8 with low silicic acid concentration, but this result doesn't meet (1). It could be explained, at least partly, by rapid selective sinking of dead populations as shown in Fig. 5. But, low % living diatoms at Stn. 4 with high silicic acid concentration was resulted from shift of equilibrium point between mortality rate and sinking rate toward

higher mortality than at the stations with high % living diatoms, again far away from (1). I am a little bit concerned about reliability of the incubation experiment because of lack of positive control (light incubation). My question is if senescence was actually induced by darkness, despite of low silicic acid concentration and difference in incubation temperature from sampling temperature. Also, I am concerned about reproducibility of the results from the sinking experiment. But, large variation in % living of aphotic diatoms is very interesting and dose it relate to selective sinking of dying/dead population? A unique feature of this study is collection of natural microphytoplankton community by the Bottle-Net, and thus I would like to suggest to conduct more detailed species-level analysis to test the hypothesis or put aside the hypothesis.

Specific comments

Incubation experiment: How did Authors get a highly active population (93.3% of % living) besides moderate % living population (average, 59.4%)?

% biomass in aphotic zone: Values in text and Fig. 4 seem not to meet the results in Table 1, if they are calculated as the ratio of Aphotic diatoms/(Aphotic diatoms + Photic diatoms), and the axis titles of Fig. 4 seem to be inverted. Please check them. But I would suggest to delete Fig. 4, because a negative correlation appears to be achieved by only one result of Stn. 4. Why was the upper sampling depth of some aphotic samples (Stns 4, 5, 7 and 8) set at deeper than 10 m below of the lower sampling depth of the upper layer? Do the terms of "upper layer", "photic layer" and "the surface layer" mean distinct depth zones?

Table 1: Chlorophyll a concentrations and mixed layer depth are valuable for understanding of the status of the study site.
* * *

---

## Author Comment (AC1) · 30 Jan 2019

Actions taken to accommodate the comments of reviewer #1.-

Reviewer#1- The manuscript describes the health status of diatoms during the course of the bloom in the Arctic. The main finding is that when diatoms are dying they sink out of the photic zone. Two main types of results are described here. First a clear and complete description of the diatoms in 8 stations around svalbard, and second an experiment testing the decay of diatoms in the dark, while comparing the sinking of living versus dead diatoms. While I feel that these data are very interesting, the findings are not new and should have been presented with others in order to give a valuable manuscript.

Authors: We thank you the reviewer for the useful comments and the time devoted to revising the manuscript. We carefully followed the reviewer's comments to improve the revised manuscript. We added more data to the manuscript as detailed below, which are now shown in the Table and in three new plots. We agree that our results are relevant, as indicated by the reviewer, but also wish to point out, that they are also original and new, as clearly stated also by Reviewer #2. There are no similar data published before, so the novelty of the results presented cannot be disputed. Whereas the patterns found here could be hypothesized or expected, such expectations cannot replace empirical demonstrations or observations. During the cruise, we used a new oceanographic device, the Bottle-net, which we described in a recent paper (Agusti et al. 2015, Nature Communications), that allows sampling of microplankton at the desired depth layers. Indeed, the system used here is advanced relative to that used by Agusti et al. 2015, and allowed sampling strategies that were not possible with the original system. Hence, no data similar to that presented here has been reported anywhere for the ocean (neither the Arctic nor anywhere else). We used this new device to sample the phytoplankton populations present in the photic and aphotic layers, separately. We obtained fresh samples from below the photic layer, and from the photic layer, and were able to test the cell health status of the cells at both layers. The number of studies quantifying diatoms health cell status of natural samples remains minimal, particularly for populations below the photic layer, which have never before been reported for the Arctic Ocean.

Action: We followed the reviewer's advice and added more data to the revised manuscript: -We included data of the upper mixed layer depth (UPM), as suggested by the reviewer, to improve the description of the environmental conditions. In pg. 3, lines 8-10, methods section we indicated: "We calculated the upper mixed layer (UPM), an index of the stability of surface water column, as the shallowest depth at which water density (sigmat) differs from surface values by more than 0.05 kg m-3 (Mura et al. 1995)". In pg. 5, lines 5-17, we added information about the UPM in the results sections, together with other environmental parameters: "The stations sampled encompassed a broad diversity of conditions, including a station where the spring bloom had not yet occurred (station 4, off the Western Svalbard shelf), as indicated by low diatom stocks and high dissolved inorganic nutrient concentrations (photic layer concentrations Si(OH)4 = 4.15 ± 0.04 $\mu$mol Si L-1, NO3 = 9.43 ± 0.09 $\mu$mol N L-1, Table 1) with lower stratification (Table 1). All other stations sampled were characterized by comparatively depleted nutrient concentrations (photic layer concentrations Si(OH)4= 0.99± 0.30 $\mu$mol Si L-1, NO3 = 1.93 ± 0.76 $\mu$mol N L-1, Table 1), thereby representing communities that were either in advanced blooming stages or were senescent after blooming. Stations 6 (SW Svalbard shelf) and 8 (E Svalbard shelf) supported actively blooming diatom populations, with the highest chlorophyll a concentration (10.5 $\mu$g Chl a L-1 for station 8), and a large fraction of living diatom cells (about 70%, Table 1). Both stations showed the highest stratification among the stations sampled, as indicated by their lower UPM values (Table 1). In contrast, Station 9 (Polar Front) supported a senescent diatom population in post-bloom phase, as indicated by depleted nutrient pools and a low percentage of living diatom cells (46.0 %, Table 1). The highest mixing was observed at the station sampled at the Barents Sea (Table 1)."

-We added a new Figure (now Figure 4), to the revised manuscript where we show the composition of the diatom community in the photic and aphotic layers. In pg. 5-6, lines 33-38, 1-4, we indicated: "The diatom community at the beginning of the cruise was dominated by Fragilariopsis spp. and Chaetoceros spp., and changed at stations 6-7-8 to communities dominated by Fragilariopsis spp. and Thalassiosira spp. that dominated the biomass where the largest diatom bloom was found (station #8, Fig. 4). Community composition changed at the Polar Front and Barents Sea stations (Fig. 4) with a larger contribution of Navicula pelagica (included in "Other", Fig. 4). The diversity of the diatoms found at the aphotic zone differed in several stations from that found at the photic layer (Fig. 4). The large celled Thalassiosira sp. colonies dominated the aphotic community in several stations although they were not dominant at the photic community (Fig. 4). At station #4, the community sampled was more diverse at the aphotic than at the photic layer (Fig. 4) indicating high sinking despite the low biomass."

- We replaced the old Figure 4 to now show two panels in the new Figure 5. Panel (a) shows the proportion (mean ± SE) for the different diatom taxa of the water-column population stock found in the aphotic zone. Panel (b) shows the relationship between the percentage of living diatoms cells in the photic layer and the proportion of the water-column population stock found in the aphotic zone for all the dominant taxa. The new figure is more informative and more significant (p< 0.001) than the previous one showing mean data values, which aggregated variability among populations. - In pg. 6 lines 4-16, in the results section, the revised text was modified to describe the new results shown, as follows: "The stock of diatoms that had sunk below the photic layer comprised, on average, 24.2 ± 6.7 % of the total water column stock, with this fraction ranging considerably between groups (Fig. 5). The proportion of biomass of the large celled Thalassiosira colonies that had sunk below the photic layer was the largest, and that of Chaetoceros spp. the smallest (Fig. 5). Station #4 in pre-bloom status showed the largest proportion of the biomass below the aphotic layer and station #8, supporting the largest diatom bloom, the lowest. At station #8, however, the population of the dominant Thalassiosira species contained 54.8 % of living cells and was paralleled with a significant contribution of dead cells at the aphotic layer (Fig. 4), suggesting the initiation of the collapse of the bloom, despite the considerable biomass standing in the photic layer. Similarly, Fragilariopsis senescence at station #3 (only 35.1 % of cells were alive at the photic layer) helps explain its larger contribution at the aphotic zone (Fig. 4). There was a significant negative relationship between the percent of the diatom stock population that had sunk below the photic layer and the percent of living cells in the photic layer ($R^2$ = 0.39, P <0.001, Fig. 5b), indicating that healthy, actively growing populations largely remain on the surface, whereas senescent ones sink out of the photic layer. "

Reviewer#1- The discussion is a little weak and rely a lot on the paper by Krause et al. For example, the discussion starts saying that diatoms in Arctic are limited by silicates and that silicates depletion is the driver of diatom death and sinking which is a result from the study by Krause et al. 2018. Why didn't you use the results of this study

regarding the survival of diatoms in the dark? Can't it be one of the trigger if the mixing increase? The paper states that the average life of the diatoms in the dark is slightly superior than a day. In this part of Arctic I guess that there is strong mixing. How long are the diatoms kept in darkness due to mixing? The data from station 9 (polar front) showed indeed that there is an effective mixing (similar diatom concentrations and % of living cells in photioc and aphotic samples), however, the % of living cells is still high. How do the authors explain that?

Authors: We revised and implemented the manuscript and the discussion in the aspects indicated by the reviewer. However, Krause et al. (which includes all of us), did not conclude that "silicates depletion is the driver of diatom death and sinking", simply because diatom death was not measured or reported in the experiments reported in Krause et al. [which were conducted at different stations as those reported here] Actions: The actions made to improve the manuscript discussion included: - Mixing conditions, as UPM included in Table 1, are now used to interpret and discuss the results at the different stations. However, mixing was not as high as suggested by the reviewer as the UPM ranged from 3 m at station 8, to 75 m at station 10. In contrast to the Southern Ocean, where mixing depths often exceed 100 m, the sector of the Arctic where we worked is characterized by shallow UPMs, as the water column is often established by ice melting or density differences between Arctic water and the underlying saltier Atlantic water. Hence, the average UPM across the study was 32.7 m, which did not extend significantly below the photic layer (average photic layer depth 40 m), implying that cells being mixed within the UPM largely experienced photic conditions. - Station #9, at the polar front, showed, however, a moderate UPM of 35 m, so we could not relate the % of living cells observed in the two layers (photic and aphotic) to mixing below the photic layer. We can however relate diatom sinking at the polar front to the bloom-stage, and to the limitation by nitrate and Si. We now include in the discussion the statement (pg. 7 lines 18-21): "A post-bloom situation was identified at the polar front community, with similar percentages of living cells at the photic and aphotic zones as a result of high sinking induced by Si and nitrogen limitation." - In relation to the dark experiments, Reviewer #2 noted that the experiments did not include a light treatment, so we could not extrapolate the decay rates solely to darkness. In the revised manuscript, we indicated that those experiments are representative of the environmental conditions in the aphotic layer, i.e darkness and other conditions, and the experiments are now referred as "aphotic conditions" instead of "darkness" alone.

Reviewer#1- The different stations are ideally located and sampled to describe the diatom bloom from the initiation to the decline, but these could be more interestingly discussed in the paper. What can be brought to light from the results of this paper? What is the bloom status at each station at the sampling time? this could be a lot more discuss using diatom cell concentrations in photic and aphotic zone, % of living cells, nutrients concentrations.... How are the nutrient concentrations compared to the winter concentrations ? that may give an idea of the bloom advancement. How is the bloom terminate? Authors: We revised this aspect in the discussion. In pg. 7 lines 10-20, the new paragraph reads: "Quantification of the % of living cells helped identify the different stages of the arctic spring bloom at the stations sampled. A pre-bloom situation, characterized by low cell abundance and a small percentage of living cells, was found at station #4, located further west off Svalbard Islands, where silicic acid and nitrogen concentrations were high and the UPM was deeper than in other arctic stations. The healthiest diatom community was observed at station #5, where the high stratification and $Si(OH)_4$ concentration above the half saturation constant (Ks) of 2 $\mu$M (from kinetic experiments in the same region by Krause et al. 2018) helped the diatoms support active growth. The highest cell abundance was observed at station #8, but the lower % of living diatoms and the $Si(OH)_4$ concentration well below the Ks value indicated that the bloom was reaching the maximum capacity, although diatom sinking was still low. A post-bloom situation was identified at the polar front community, with similar percentages of living cells at the photic and aphotic zones as a result of high sinking induced by Si and nitrogen limitation."

Reviewer#1- Why these data are not in the paper by Krause et al if it uses so much

of the conclusions issued from it? Alone I feel that these data even if very interesting are too poor. Authors: In the manuscript, we reported original data based on the new methodology, and both the goals addressed and the results obtained are not the same as those described on the manuscript by Krause et al. As indicated above, we included more data in the revised manuscript and we followed the reviewer suggestions and improved the discussion to deviate from Krause et al. manuscript discussion on Si limitation. Note, that the stations sampled in Krause et al. and those we sampled often did not match due to operational limitations of cable time and water budgets available, so Krause et al. used a sampling and experimental strategy completely different from that used here (as well as variables and processes resolved). Hence, any attempt to combine Krause et al. results, which focus on Si uptake kinetics resolved through experimental additions of Si, with those presented here would have been lead to high inconsistencies. We used the conclusions by Krause as a starting point, whereas our conclusions are self-standing and do not depend on results presented in Krause et al.

Action: As indicated above, we added more data and detail on the community composition described in three new plots (new Figs 4 and 5).

Reviewer#1- What are the limitations there? Why do the authors state that there is only silicate limitations and not nitrate while nitrate are also very depleted in some zone (station 6, 7 and 8) Authors: We revised the manuscript to increase clarity on this aspect. The high requirements of diatoms for Si imply that silicon limitation could led to diatom bloom collapse before nitrogen would be exhausted. Kinetic experiments by Krause et al 2018), indicated that the half saturation constant (Ks) of $Si(OH)4$ was above of 2 $\mu$M (from kinetic experiments in the same region by Krause et al. 2018) for most communities which was above the $Si(OH)4$ concentration in the water. In any case, we revised this aspect in the manuscript because other drivers, as mixing and other nutrients (nitrogen), would contribute to the variability described in the study, and we now acknowledge the role of depleted nitrate pools as well. Action: The actions made included: -In the abstract, pg 1, lines 35-37. We corrected the paragraph that

now reads: "The results conform to a conceptual model where diatoms grow during the bloom until resources are depleted, and support a link between diatom cell health status and sedimentation fluxes in the Arctic". - In pg 7 lines 18-20, we modified the paragraph as follows: "A post-bloom situation was identified at the polar front community, with similar percentages of living cells at the photic and aphotic zones as a result of high sinking induced by Si and nitrogen limitation, as suggested by the lower Si(OH)4 KS of 0.8 $\mu$M (Krause et al. 2018). " -pg 7, lines 24-30. We modified the paragraph at the end of the discussion as follows: "When compared across the contrasting stages of bloom development represented in the data set analyzed here, the results presented conform to a conceptual model where nutrients, including Si (Rey 2012; Krause et al., 2018), and mixed layer drives the growth of diatoms during the Arctic spring bloom (Wassmann et al., 1997; Reigstad et al 2002). For diatoms, Si depletion results in two potential physiological issues: yield limitation (i.e. diatom standing stock is too high to be supported by the available silicic acid) and intense kinetic/growth limitation (i.e. depleted silicic acid silicic acid limits diatom Si uptake to such a degree that growth must slow, Krause et al., 2018)".

Reviewer#1- It would have been great to discuss them in light with production rates, limitations or sinking fluxes of bSi or POC from sediment traps data. Authors: We agree that these comparisons would be relevant, but despite our great interest, these data sets did not match due to logistic requirements of the operation of the Bottle-nets and CTD sampling and sediment trap operations, so these data sets are largely disjoint for the cruise, with measurements conducted in different stations. This is, as explained above, one of the rationales why these results and those reported in Krause et al. (2018) could not be integrated onto a single paper. For example, the number of sediment traps deployed was low, only two of them were deployed in the same area sampled by Bottle-Nets (Hornsund and Erik Eriksen strait), but not at the same position and were deployed on a Lagrangian, drifting, mode, with the depths of deployment more shallower than the stations, further offshore, where bottle nets were deployed. In any case, in the revised version we now refer to results obtained by the sediment traps

deployments during the study (reported in Krause et al. 2018).

Action: In the revised manuscript, at pg. 7, lines 20-23 we added the following paragraph: "The diatom community captured by the bottle net below the photic layer was consistent with the limited but comparable data obtained with results obtained from sediment traps deployed in the area, which also indicated Fragilariopsis and Thalassiosira species to be the dominant contributors to Si and biomass export (Krause et al. 2018). "

New references: Mura, M.P., M. P. Satta and Agustí, S.: Water-mass influences on summer Antarctic phytoplankton biomass and community structure, Polar Biology, 15 (15-20), 1995.

Reigstad, M., Wassmann, P., Riser, C. W., Øygarden, S., and Rey, F.: Variations in hydrography, nutrients and chlorophyll a in the marginal ice-zone and the central Barents Sea, J. Marine Syst., 38, 9–29, 2002.

New Figures are copied below

───────────────────────────

[Figure]

**Fig. 1.** New Figure 4: Pie charts showing the diatom community at the photic and aphotic zones. The colors correspond to different taxa

(a)

[Figure]

(b)

[Figure]

**Fig. 2.** New Figure 5: with new plots (a) and (b)

[Figure]

---

## Author Comment (AC2) · 30 Jan 2019

Actions taken to accommodate the comments of reviewer #2.-

Reviewer#2- This MS sheds a light on the role and fate of diatoms over a course of a spring bloom in the Arctic Ocean, based on the estimates of their mortality, senescent rate, and the population with fast sinking rate. These estimations were designed to test a hypothesis in which Si-depletion triggers (1) senescence of diatoms and (2) selective sinking of the dying population. Because of intense $CO_2$-sequation in the Arctic Ocean, this hypothesis is valuable to be tested, but the results in this study unlikely support this hypothesis.

Authors: We thank you the reviewer for the useful comments and the time devoted to

revising the manuscript. We agree that the results presented are limited in terms of testing the hypothesis of a direct relationship between the percentages of living cells, whether found at the photic layer or exported, with Si-depletion, as a direct link with Si depletion can be suggested, but not demonstrated, since nitrate levels were also low when Si was depleted (as also pointed out by rev. #1). Instead, our study provides a more reliable test of hypothesis (2). We have now revised this manuscript to focus on hypothesis (2), while more broadly suggesting that nutrient – not exclusively Si – depletion leads to senescence of diatoms. As a general comment we also outline the inherent difficulties of addressing questions on diatom blooms in the Arctic that require direct sampling. Ship time is typically secured 2 years ahead and there is no margin to accommodate to the nuances encountered every year, which involve different phenology of the blooms and unpredictable seaice conditions. Hence, such cruises need be adaptive, more so because the goals of all other teams sharing ship time are adaptive themselves. Conducting such studies in polar waters, on which we are highly experienced (both Arctic and southern Ocean), involves, therefore, considerable doses of contingency. For instance, the reviewer raises, rightly so, concerns on the reliability of the experiments, since often a single experiment was conducted. We would have liked to conduct many more experiments, but this was precluded by operational reasons. We have, thus, toned down the conclusions derived from the experiments, and used them more as supportive evidence for the collective insights derived from the entire set of measurements, rather than stand-alone evidence. Action: We modified those paragraphs related to hypothesis (1) to increase clarity, as follows: -In the abstract, pg 1, lines 35-37. We corrected the paragraph that now reads: "The results conform to a conceptual model where diatoms grow during the bloom until resources are depleted, and support a link between diatom cell health status and sedimentation fluxes in the Arctic." -pg. 7, lines 24-30. We modified the discussion as follows: "When compared across the contrasting stages of bloom development represented in the data set analyzed here, the results presented conform to a conceptual model where nutrients, including Si (Rey 2012; Krause et al., 2018), and mixed layer drives the growth of diatoms during the Arctic spring bloom (Wassmann et al., 1997; Reigstad et al 2002). For diatoms, Si depletion results in two potential physiological issues: yield limitation (i.e. diatom standing stock is too high to be supported by the available silicic acid) and intense kinetic/growth limitation (i.e. depleted silicic acid silicic acid limits diatom Si uptake to such a degree that growth must slow, Krause et al., 2018)." - and in pg. 8, lines 28-29 : "Deterioration of diatom health, such as occurring when reaching acute silicon or other resources limitation along the spring bloom,…".

Reviewer#2 For example, high % living diatoms in the upper layer was achieved at Stns 6, 7 and 8 with low silicic acid concentration, but this result doesn't meet (1). It could be explained, at least partly, by rapid selective sinking of dead populations as shown in Fig. 5. But, low % living diatoms at Stn. 4 with high silicic acid concentration wasn resulted from shift of equilibrium point between mortality rate and sinking rate toward higher mortality than at the stations with high % living diatoms, again far away from (1).

Authors: We agree that the results presented do not suffice to identify Si limitation; a diagnosis of whether Si limits diatom production should be accompanied by additional analyses and experimental additions. In the manuscript of Krause et al. 2018, kinetic data during the same cruise indicated that in three of four experiments KS (half-saturation constant for Si(OH)4) was approximately 2.0 $\mu$M, indicating that Si was already exhausted in the stations showing the higher biomasses. In the Polar Front we observed a situation of post-bloom, and Ks there was found to be lower. Action: We revised the manuscript and modified the text in the discussion, and more broadly referred to nutrient, rather than just silicon, limitation. We added a paragraph in the discussion, indicating the situation at the different stations sampled, concerning the environmental conditions found including mixing (as suggested by reviewer #2) and the health status of the cells: - In pg. 7 lines 10-20, the new paragraph reads: "Quantification of the % of living cells helped identify the different stages of the arctic spring bloom at the stations sampled. A pre-bloom situation, characterized by low cell abundance and a small percentage of living cells, was found at station #4, located further west off Svalbard

Islands, where silicic acid and nitrogen concentrations were high and the UPM was deeper than in other arctic stations. The healthiest diatom community was observed at station #5, where the high stratification and Si(OH)4 concentration above the half saturation constant (Ks) of 2 $\mu$M (from kinetic experiments in the same region by Krause et al. 2018) helped the diatoms support active growth. The highest cell abundance was observed at station #8, but the lower % of living diatoms and the Si(OH)4 concentration well below the Ks value indicated that the bloom was reaching the maximum capacity, although diatom sinking was still low. A post-bloom situation was identified at the polar front community, with similar percentages of living cells at the photic and aphotic zones as a result of high sinking induced by Si and nitrogen limitation."

Reviewer#2.- I am a little bit concerned about reliability of the incubation experiment because of lack of positive control (light incubation). My question is if senescence was actually induced by darkness, despite of low silicic acid concentration and difference in incubation temperature from sampling temperature. -Authors: We agree that the incubations could inform on the mortality when reaching the aphotic zone, but do not represent the response to "darkness" due to the lack of a parallel light control. Action: We modified the text to reduce the emphasis on "darkness" and clarify that those incubations may represent the response to the environmental conditions below the photic layer, that involve darkness and other changes. In pg 6, lines 25-26: "The experiment testing diatom survival in aphotic zone light conditions conducted indicated that once diatom cells sink below the photic layer, they would die rapidly." In pg 7, lines 4-6: "Moreover, our experimental assessment of diatom survival incubated at aphotic conditions suggested that once sinking below the photic layer, diatoms cells could die at half-lives of 21.8 to 30.2 hours across species." In pg 12, in the Figure 6 heading: "Decay in the cell abundance of living (blue diamonds) and total cells (orange squares) of arctic diatoms when exposed to aphotic zone light conditions."

Reviewer#2.- Also, I am concerned about reproducibility of the results from the sinking experiment. But, large variation in % living of aphotic diatoms is very interesting

and dose it relate to selective sinking of dying/dead population? A unique feature of this study is collection of natural microphytoplankton community by the Bottle-Net, and thus I would like to suggest to conduct more detailed species-level analysis to test the hypothesis or put aside the hypothesis. -Authors: We agree that more sinking experiments will be convenient, but we were not able to duplicate the sinking experiment because the column was used by the zooplankton group for sampling marine snow, and our experiment required more than 48 hours to be completed. Provided we present a single experiment, we have toned down the conclusions and use the experiment as an additional source of evidence, rather than a conclusive demonstration on its own right. - We agree with the reviewer that the presentation of results from the experiment we were able to conduct would benefit from adding more detailed information at the species level in the results. Reviewer #1 also suggested to add more detailed results, and we added more detailed data in the revised manuscript at the taxonomic level. Action: -We added a new Figure to the revised manuscript where we show the composition of the diatom community in the photic and aphotic layers. This is the new Figure 4, in the revised manuscript. In pg. 5-6, lines 33-38, 1-4, we indicated: "The diatom community at the beginning of the cruise was dominated by Fragilariopsis spp. and Chaetoceros spp., and changed at stations 6-7-8 to communities dominated by Fragilariopsis spp. and Thalassiosira spp. that dominated the biomass where the largest diatom bloom was found (station #8, Fig. 4). Community composition changed at the Polar Front and Barents Sea stations (Fig. 4) with a larger contribution of Navicula pelagica (included in "Other", Fig. 4). The diversity of the diatoms found at the aphotic zone differed in several stations from that found at the photic layer (Fig. 4). The large celled Thalassiosira sp. colonies dominated the aphotic community in several stations although they were not dominant at the photic community (Fig. 4). At station #4, the community sampled was more diverse at the aphotic than at the photic layer (Fig. 4) indicating high sinking despite the low biomass." - We changed the old Figure 4 to show a new Figure 5, with two panels. Panel (a) shows the proportion (mean ± SE) of the water-column population stock found in the aphotic zone for the different diatom taxa.

Panel (b) the relationship between the percentage of living diatoms cells in the photic layer and the proportion of the water-column population stock found in the aphotic zone but for all the dominant taxa. The new figure is more informative and highly significant ($R^2$ of 0.39 and p< 0.001). - In pg. 6 lines 4-16, the revised text was also modified as follows: "The stock of diatoms that had sunk below the photic layer comprised, on average, $24.2 \pm 6.7$ % of the total water column stock, with this fraction ranging considerably between groups (Fig. 5). The proportion of biomass of the large celled Thalassiosira colonies that had sunk below the photic layer was the largest, and that of Chaetoceros spp. the smallest (Fig. 5). Station #4 in pre-bloom status showed the larger proportion of the biomass below the aphotic layer and station #8, supporting the largest diatom bloom, the lowest. At station #8, however, the population of the dominant Thalassiosira species contained 54.8 % of living cells and was paralleled with a significant contribution of dead cells at the aphotic layer (Fig. 4), suggesting the initiation of the collapse of the bloom despite the considerable biomass standing in the photic layer. Similarly, Fragilariopsis senescence at the photic layer of station #3 (only 35.1 % of cells were alive at the photic layer) helps explain its larger contribution at the aphotic layer (Fig. 4). There was a significant negative relationship between the percent of the diatom stock population that had sunk below the photic layer and the percent of living cells in the photic layer ($R^2 = 0.39$, P <0.001, Fig. 5b), indicating that healthy, actively growing populations largely remain in the surface, whereas senescent ones sink out of the photic layer. "

Specific comments

Reviewer#2.- Incubation experiment: How did Authors get a highly active population (93.3% of % living) besides moderate % living population (average, 59.4%)? Authors: We agree that the information was presented in a confusing manner. It was provider in the methods section and it is the mean corresponding only to the two dominant species. The communities were sampled at Erik Eriksen Strait where the % living cells of 70% was higher than the cruise average of 59.4%. Action: We removed this information

from the methods section to avoid confusion.

Reviewer#2.- % biomass in aphotic zone: Values in text and Fig. 4 seem not to meet the results in Table 1, if they are calculated as the ratio of Aphotic diatoms/(Aphotic diatoms + Photic diatoms), and the axis titles of Fig. 4 seem to be inverted. Please check them. But I would suggest to delete Fig. 4, because a negative correlation appears to be achieved by only one result of Stn 4. Authors: The original Figure 4 showed the average values obtained for the dominant species at each station. This explains the mismatch observed by the reviewer between the data in Table 1 and those in Figure 4. Action: We revised and reorganize this information for consistency. Action: As indicated above, we modified Figure 4 in the revised version of the manuscript, showing now the relationship of the dominant diatom groups (new Figure 5). This relationship is stronger and is based on a larger number of data. We also revised and corrected some typos in the Table.

Reviewer#2.- Why was the upper sampling depth of some aphotic samples (Stns 4, 5, 7 and 8) set at deeper than 10 m below of the lower sampling depth of the upper layer? Authors: Those stations were strongly stratified as observed in the CTD profiles of fluorescence and light, and 10 m separation was enough to perfectly separate the sampling of the two layers to ensure samples did not overlap.

Reviewer#2.- Do the terms of "upper layer", "photic layer" and "the surface layer" mean distinct depth zones? Action: We agree, and have revised the manuscript to used "photic" throughout.

Reviewer#2.- Table 1: Chlorophyll a concentrations and mixed layer depth are valuable for understanding the status of the study site. Action: We calculated and added data of the upper mixed layer (UPM) for each station in the revised Table 1. We do not have the data of chlorophyll a concentration for all the stations, as this was not analyzed for all the stations. We provide the data on the abundance of cells, as it is a good indicator of the phytoplankton biomass at each station, and also add the range in Chla values

obtained during the study in the results section.

New plots are copied below

[Figure]

[Figure]

**Fig. 1.** New Figure 4: Pie charts showing the diatom community at the photic and aphotic zones. The colors correspond to different taxa

(a)

[Figure]

(b)

[Figure]

**Fig. 2.** New Figure 5: with new plots (a) and (b)

---

## Referee Report (RR1)

Revised MS has been improved to describe the Author's logic more clearly along the line of (1) nutrient depletion leads unhealthy status of diatom --> (2) unhealthy/senescent diatom sinks down faster than healthy diatom --> (3) diatoms reaching the aphotic zone die and lyse soon --> (4) downward export of unhealthy/senescent diatom increase the progression of the surface pCO2 unsaturation. Results from the experiments fully support (2) and (3), but not (1), even though many previous studies suggest it. I would like to request Authors (i) to state insignificant correlation between nutrient concentrations and %living biomass in the photic zone, and, (ii) if possible, to cite the previous studies on the mechanisms driving diatom senescence other than nutrient depletion. My other concern is about (4), as rapid lysis of senescent diatoms (3) might result in supply of suspended/dissolved organic carbon and their remineralization within the subsurface waters which reduces air sea disequilibrium in the surface through processes such as upwelling, mixing and vertical diffusion (Bates and Mathis, 2009). I suggest Authors (iii) to explain the contradicting effects of (2) and (3) for the surface pCO2. There are many errors in writing. I recommend (iv) to ask an English proofreading service before re-submission. I hope my comments would help Authors to enhance the value of their findings.

Specific comments

p. 4, l. 26: Write the station ID (#3?) whose sample was used in the sinking experiment.

p. 5, l. 7: NO3 should be NO3 + NO2.

p. 6, l. 2 – 4: "At station #4, the community sampled was more diverse at the aphotic than at the photic layer (Fig. 4) indicating high sinking despite the low biomass." Why does high sinking rate contribute to higher diversity in the aphotic zone than that in the photic zone?

p. 6, l. 17 – 16: "Initially, only 6.7 % of the cells of the Flagilariosis sp. and Thalassosira sp. colonies dominating the community tested were dead." There seems no stations where these two species were dominant in Fig. 4. Again, at which station was the sinking experiment conducted?

p. 6, l. 29 – 30: "6.3 days for the largest Thalasiosira sp. cells" "6.3 days" would be "5.3 days".

p. 7, l. 18: "..., although diatom sinking was still low. " Was it confirmed at this station?

---

## Referee Report (RR2)

The article studies the impact of life stage of diatoms on their vertical export in the Svalbard area. The article would confirm the hypothesis that unhealthy diatoms are sinking whereas the healthy ones maintain better buoyancy and rather stay at surface.

I quite enjoyed the reading and I found it very instructive.

First, the topic is highly strategic and interesting. Diatoms dominate the primary production in this very productive region. However, very few (or no?) studies have in situ measurements. The hypothesis that senescent diatoms sink is often use without proper in situ proof. For these reasons, I think this article could potentially provide a valuable contribution.

However, I could just agree with the other reviewers. I have concerns about the significativity of the results which are based on a very limited dataset. Yes, the authors found significative differences between photic and aphotic communities but still, this is a very weak. I acknowledge that the authors have done a great effort to re-frame their results during the review process and that there is not much more information to extract from the data set. I recommend say 'moderate' revisions to address remaining issues.

About the rate of mortality, I agree with most of the authors answer to reviewer #2. But I deeply encourage authors to be more specific. For example, it was not clear, before I read the answer to reviewer #2 the subtilty in the terminology of "survival". Please clarify that you only consider vegetative cells here. Because this is clear there is survival of (resting) cells in the dark for a long period of time.

Perhaps it could serve also in the discussion to understand that your approach is different from other studies. I have to admit I had exactly the same reaction than reviewer #2 during my first read of the revised manuscript (first appeared contradictory with the literature I know)

I was also surprised some major references were not listed in your study. As global critic, this article still lacks of contextualization. Some articles (see below in gray) that I found related (you are not obliged to cite those but it could help you to improve the discussion).

See for example

 Kvernvik et al. 2019, https://doi.org/10.1111/jpy.12750. They say for example: "Our results suggest that some Arctic autotrophs maintain fully functional photosystem II and downstream electron acceptors during the polar night… This could allow Arctic microalgae to endure the polar night without the formation of dormant stages, enabling them to recover and take advantage of light immediately upon the suns return during the winter–spring transition.". I am really surprised the authors did not take more precaution while discussing this still debated topic.

In Berge et al. 2015:

"Many Arctic phototrophic plankters are able to persist during unfavorable conditions as resting stages such as spores or cysts (Garrison, 1984; Smetacek, 1985; Krempand Anderson, 2000), and diatoms are known for their potential to survive long periods of darkness (Antia and Cheng, 1970; Smayda and Mitchell-Innes, 1974; Palmisano and Sullivan, 1982; Sakshaug et al., 2009; Quillfeldt et al., 2009). The survival strategies of the various plastidic flagellates of Arctic waters throughout the dark period, however, are largely unknown."

Lacour et al. (2019) also suggest the opposite (https://link.springer.com/article/10.1007%2Fs00300-019-02507-2))
"Chaetoceros neogracilis was not able to grow in the dark but cell biovolume remained constant after 1 month in darkness. Rapid resumption of photosynthesis and growth recovery was also found when the cells were transferred back to light at four different light levels ranging from 5 to 154 µmol photon m−2 s−1. This demonstrates the remarkable ability of this species to re-initiate growth over a wide range of irradiances even after a prolonged period in the dark with no apparent lag period **or impact on survival**."

As well as the extensive synthesis by Wulf et al. 2008 (https://www.tandfonline.com/doi/abs/10.1080/0269249X.2008.9705774)
"Based on the rapid increase in Fv/Fm we safely draw the conclusion that although the cells probably were physiologically resting in the dark they were not forming resting stages such as spores or cysts. Physiologically resting cells are morphologically similar to the vegetative cells, but are physiologically dormant and can be induced when cells are transferred to cold and dark conditions (Anderson 1975a). Like in our study, these cells have condensed protoplasts which are transformed back to the former state (within hours) upon re-exposure"

How the authors can be 100% confident that their unique culture experiment was reliable? Did something else than darkness could have killed the algae (i.e. contamination?) ? Could you discuss that ?

Also, there is many syntax, grammar and terminology issues in the text (lot of them were introduced during the revision process, that is a pity). I recommend to carefully correct the text because it makes the understanding sometimes difficult (I had to re-read many sentences many times). I have tried to list them in the specific comments but it became quickly overwhelming.

I encourage the authors to continue their efforts.

Specific comments:

PAGE9 (abstract).
L.23: specify phytoplankton bloom, it could be sea ice algae. Or if no distinction, just microalgae.
L.26: regional = specify Svalbard or Northwest Barents Sea
L.27-28: very awkward sentence. Change "occurrent with" something like "together with".

L.29: SE= Standard Error ? I don't think you could use undefined abbreviation in the abstract. Need confirmation from editor.

PAGE11 (2.1)
L.23: you canno't say MLD is an indicator of stability. Stability is usually related to stratification. Stratification is basically the density gradient which you don't measure here. MLD is an indicator of mixing, this is it. People know what is MLD (or UPM), I suggest you just erase "an index of the stability of the water column" and also later in the text.

PAGE12 (2.3)
L.32 Move "expected" before "mortality".
L.35: I can understand but this is not well written. Perhaps change ", this simulated" par "simulating" ? or "which simulated" ?

PAGE13 (3)
Please be consistent and use either r or $R^2$ throughout the manuscript.

PAGE14
L.10: N or n ?
L.13: "from station 6 to station8"
L.14: a E is missing in the verb were. Change "these wre also the areas with" by something like "and where"

L.31-32: this is interpretation, should be moved in discussion.

PAGE 15, LINE 5 and PAGE 16 LINE 7: please do not use the term trend in this context. This is not appropriate, please re-word.

PAGE16
Line 23: amoung out ??? amoung OUR ?
Line 35-37: weird wording, had to re-read several times. I guess you wanted to say sedimentation is enhanced BY higher quotas for polar diatoms. Please re-word.

---

## Author Response (AR2)

**Actions taken to accommodate the 2nd revision comments of reviewer #1 on "Arctic (Svalbard Islands) Active and Exported Diatom Stocks and Cell Health Status"** by Susana Agustí et al.  https://doi.org/10.5194/bg-2018-459-RC1, 2018

5   Reviewer #1.- Revised MS has been improved to describe the Author's logic more clearly along the line of (1) nutrient depletion leads unhealthy status of diatom --> (2) unhealthy/senescent diatom sinks down faster than healthy diatom --> (3) diatoms reaching the aphotic zone die and lyse soon --> (4) downward export of unhealthy/senescent diatom increase the progression of the surface pCO2 unsaturation. Results from the experiments fully support (2) and (3), but not (1),

10   even though many previous studies suggest it. I would like to request Authors (i) to state insignificant correlation between nutrient concentrations and %living biomass in the photic zone, and, (ii) if possible, to cite the previous studies on the mechanisms driving diatom senescence other than nutrient depletion. My other concern is about (4), as rapid lysis of senescent diatoms (3) might result in supply of suspended/dissolved organic carbon and their remineralization

15   within the subsurface waters which reduces air sea disequilibrium in the surface through processes such as upwelling, mixing and vertical diffusion (Bates and Mathis, 2009). I suggest Authors (iii) to explain the contradicting effects of (2) and (3) for the surface pCO2. There are many errors in writing. I recommend (iv) to ask an English proofreading service before re-submission. I hope my comments would help Authors to enhance the value of their findings.

*Authors:* We thank the reviewer for the useful comments.  We implemented the revised version with the suggestions of the reviewer.
*Action:* We added different paragraphs and corrections as indicated below::
(i) p. 5 Lines 26-27: *"The percentage of living cells was not significantly correlated with the*

25   *concentration of $NO_3 + NO_2$ (2-tail test, r = -0.54, P = 0.17) or $Si(OH)_4$ (2-tail test, r =-0.69, P = 0.06)."*
(ii) p. 2 Lines 23- 25, we added the paragraph: "*Alou-Font et al. (2016) observed large variability in the health status of phytoplankton in the Canadian Arctic, influenced by the light and temperature conditions, but not by nitrate concentration —typically thought to be the main*

30   *yield-limiting nutrient.*"
In p. 8, we added the following paragraph: *"Cell abundances and health state observed were consistent with previous studies.  In the Canadian Arctic, living cells in open-water and ice-covered stations represented the $57.3 \pm 5.8\%$ and $48.0 \pm 3.9\%$ ($\pm$ SE), respectively (Alou–Font et al., 2016), similar to proportions in our study. The percentage of living cells was higher during*

35   *the bloom periods than periods before and after (Alou-Font et al. 2016).  This trend appeared to be driven by light and low nutrient concentrations (Alou–Font et al., 2016)."*
(iii) There is no contradiction because dead cells sink below the photic layer. Also, because phytoplankton cell mortality and lysis imply several steps of cell degradation (e.g. Segovia et al. 2003), the release of the carbon will follow the degradative process. If dead cells sink below the

40   photic layer most of the carbon will be released below the photic layer.  In p.8 Lines 25-29, we added the following paragraph: "*Acute silicic acid limitation, is identified, therefore, as the event leading to loss of the capacity to actively regulate buoyancy that characterizes diatom cells (Smayda, 1970), and rapid sinking of the bloom.  The potential for rapid sedimentation is enhanced, due to the higher silica quotas for polar diatoms (Lomas et al. 2019) compared to*

45   *lower-latitude diatoms (Brzezinski 1985)."*
(iv)  English native coauthors revised the language.

Reviewer #1.   Specific comments
p. 4, l. 26: Write the station ID (#3?) whose sample was used in the sinking experiment.
p. 5, l. 7: NO3 should be NO3 + NO2.
p. 6, l. 2 – 4: "At station #4, the community sampled was more diverse at the aphotic than at the photic layer (Fig. 4) indicating high sinking despite the low biomass." Why does high sinking rate contribute to higher diversity in the aphotic zone than that in the photic zone?
p. 6, l. 17 – 16: "Initially, only 6.7 % of the cells of the Fragilariopsis sp. and Thalaisiosira sp. colonies dominating the community tested were dead." It seems there is no stations where these two species were dominant in Fig. 4. Again, at which station was the sinking experiment conducted?
p. 6, l. 29 – 30: "6.3 days for the largest Thalaisiosira sp. cells" "6.3 days" would be "5.3 days".
p. 7, l. 18: "..., although diatom sinking was still low. " as it confirmed at this station?

*Authors:* We corrected all the specific comments in the revised manuscript.
* * *
**Actions taken to accommodate the 2$^{nd}$ revision of reviewer #2 on "Arctic (Svalbard Islands) Active and Exported Diatom Stocks and Cell Health Status"** by Susana Agustí et al.
https://doi.org/10.5194/bg-2018-459-RC1, 2018

Reviewer #2. Why I agree that bottle net, is a very exciting and original tool, the results here are still too poor to be used alone or to state that silicon is the trigger of the bloom senescence and sinking.
I disagree with the authors when they state that they include a sufficient amount of data to strengthen this paper. To my point of view, the paper rely too much on two experiments that were not robust enough to conclude anything, especially for the one where we could find some opposite results in the literature (which is not discussed at all). At the minimum this paper should discuss the problem with the experimental set up.

*Authors.-* The major contribution of the manuscript is the innovative relationship between the health status of diatoms in the photic layer and the large stock of diatoms exported below this layer when resource-depletion leads to poor diatom health state in the photic layer, delivered from the use of the bottle net.  In contrast, the experiments presented have simply a supporting role, but are not central to the argument.

The argument that the data set does not allow to conclude that silicon is the trigger of the bloom senescence and sinking was already addressed and was removed from the previous version, so we are surprise the reviewer continues to criticize such argument, which is no longer presented.

We agree that the experiments ran on board are limited in number but are, however, a relevant contribution examining both processes, sinking and cell death (under aphotic conditions), on

natural communities at conditions close to '*in situ*". The work on board a research vessel has limitations, particularly so in the Arctic where changing conditions impose changing cruise plans, but offers the opportunity to evaluate responses of natural communities at '*in situ'* conditions. Most literature data on the topic came from laboratory axenic cultures growing in
5      optimized conditions, that do not exist at the sea.

We agree, indeed, that the experimental results should be compared to other evidence present in the literature, which we now do.  However, the evidence present in the literature is consistent with our results, not opposite as the reviewer asserts:

Our results showing mortality rates in the dark of natural arctic diatoms populations are not in conflict with the literature.  Vegetative cells have been shown to be unable to survive in darkness (e.g., Figure 1 below from Segovia et al. 2003) and only populations producing resting stages, e.g. spores, survive.  We note, however, that not all diatoms are able to produce resting stages.
15     We provide a more detailed assessment of the experimental evidence in response to the comment on this experiment below, and we have now added a paragraph in the discussion comparing our results with those of other experiments in the discussion section.

20     Reviewer #2. Diatom survival to darkness:
I am still not convince by this experiment that aim at measuring the survival of diatom in the dark…
First, if I understood correctly it has only be done once and only at St 3.
*Authors.-* We indicated in the methods that information: the experiment was made with the
25     plankton community sampled at station #3.
Reviewer #2.  What was the health status of the cells at the start of the experiment?
*Authors.-* It was a reasonably healthy community (the % of living cells, 63.64 %, was already indicated in Table 1).

30     *Action.-* We added more information to contextualize this experiment (p 7, starting line 16).

Reviewer #2.  Second, I am not sure that the experimental protocol was sufficiently robust to conclude anything from it.

35     *Authors.-* The experiment aimed to test natural community cell death at darkness as found by phytoplankton when transiting to the aphotic layer.  Our experiment is a valuable test because most data published on the topic (including those papers cited by the reviewer) are from "*in vitro*" assays ran in the laboratory, under conditions far from those found at sea: monospecific cultures growing in culture media with high nutrients, axenic, and other optimized conditions,
40     and able to produce resting stages.
*Action.-*  As stated above, try to add more context as to the initial condition of the cells in this experiment.

Reviewer #2.  Third, the results are not at all compared to other study where the authors stated
45     something totally different from a more robust experimental set up:
Why are these results so different from the one by Smayda and Mitchell-Innes 1974 showing survival of diatoms for 90 days in the dark?

And by the study by Lewis et al. 1999 where some diatoms even survive up to 112 month?

*Authors.-* We disagree with the reviewer assessment. Careful reading of the papers the reviewer cites show that our results showing arctic diatoms mortality rates in the dark are not in conflict with the literature.

First, some of the papers the reviewer cite do not focus on the survival/mortality of the phytopklankton cells in the community, but on the capacity of resting stages, often sampled from sediments, to germinate and give rise to a population when placed under light, which can hardly be compared to the experiments we report. In previous studies different authors identified that vegetative cells could not survive in darkness, and populations must produce resting stages:

Smayda and Mitchell-Innes (1974), and Lewis et al. (1999) experimentally assessed the capability of resting stages of phytoplankton to resume growth after placement in light, which they referred to as "survival". Note that we use "survival time" to refer to the time for the population of vegetative cells to be reduced to half under darkness, which is very different from the meaning of "survival" in those experiments (i.e. it suffices for one resting stage to become viable in their experiments for a population to be developed when transferred to the light), but this is no way in conflict with our results.

Moreover, Lewis et al. 1999, did not work with phytoplankton communities, they collected cysts from sediments and germinated in the lab. Cysts are morphological and physiologically different from vegetative phytoplankton cells, and the finding that some cysts from sediments can be germinated is not at all in conflict with results showing high mortality rates of vegetative cells under darkness.

However, Smayda and Mitchell-Innes (1974) also evaluated the mortality of vegetative cells of *Chaetocesos curvissetum* under darkness, which can indeed be compared to our experiment. We, therefore, recalculated the corresponding mortality rates from the data Smayda and Mitchell-Innes (1974) provided to yield a mortality rate in darkness of vegetative *Chaetocesos curvissetum* cells of 0.19 d$^{-1}$, very close to the rates we found for "natural diatom populations" in our paper. Hence, the comparable experiment presented by Smayda and Mitchell-Innes (1974) is consistent with our findings.

A more thorough assessment of the recent literature reveals that Segovia et al. 2003 and Segovia and Berges 2009 described the he processed involved in phytoplankton cell death in monospecific cultures under darkness, using molecular tools and/or differentiate living from dead vegetative cells, concluding that vegetative cells are unable to survive in the dark. The photograph below (Segovia et al. 2003, Fig. 1), is compeling as to the outcome of exposure of the culture to 8 days of darkness.

[Figure]

**Fig. 1.**    Download figure | Open in new tab | Download powerpoint

Effects of light deprivation in *D. tertiolecta*. A, Actively growing culture. B, Culture after 8 d in darkness.

Agusti et al. (2015) is, to the best of our knowledge, the only paper reporting mortality rates in the dark of natural (subtropical) phytoplankton communities. The rates reported (from 0.06 to 0.27 d$^{-1}$) are also consistent with those we report for the Arctic community studied.

Provided these results, we find no basis on the reviewer argument that our results are inconsistent with the literature.  They are indeed perfectly consistent.

*Action:*  We added a discussion of previously reported experiments and cited the literature on the topic. In p 7, starting line 20, we added the following paragraph: *"Smayda and Mitchell-Innes (1974) also reported the decrease in viable cells after darkness: "After 6 days of dark incubation, the number of viable cells of Chaetoceros curvisetus recognizable decreased from 760 to 240 cells per ml", representing a decay rate of 0.19 d$^{-1}$ (i.e. 50% loss of cells in 3.6 days) comparable to the rate reported here. Other studies also reported cell mortality in darkness (Segovia and Berges 2009; Agusti et al. 2015), also yielding mortality rates close to those reported here.  Phytoplankton vegetative cells do not survive under darkness (Smayda and Mitchell-Innes 1974; Segovia and Berges 2009; Segovia et al. 2003) and only resting spores or resting cells are able to survive in the dark (Ignatiades and Smayda 1970; Smayda and Mitchell-Innes 1974; Peters and Thomas 1996)."*

We also added more information in the methods section and in the discussion to highlight the relevance of this experiment.

In p 7, lines 18-19: *"This result, although limited to a single experiment, was consistent among the major genera and functional groups analyzed and reflected survival at "in situ" conditions in the aphotic layer."*

Reviewer #2. How the same cells would survive in the same conditions but with light?

*Authors.-* As we indicated in the previous revision, our aim was not to test the survival of the phytoplankton on the photic layer, but to assess what is their likely fate once they fall below the photic layer. Whether they grow, remain stable or decay in the light is not relevant to understanding what is their fate in the aphotic layer.

5 *Action:* We already specified this aspect in the previous revised manuscript in response to the reviewer's comments, but have modified the statement of the aim of the experiments in the Methods section as (p4, lines 27-29): "*The mortality rates of living phytoplankton cells expected when transferred from the photic to the aphotic layer were examined at station #3 with vertical tows from the photic layer.* ".

Reviewer #2. What was the cell concentration in the experiment?
*Action:* In p 6, last paragraph, we added the following information: "*The incubation was performed close to the temperature below the photic layer that averaged 2.978 °C ± 0.003 at station #3, suggesting that trends were not driven by thermal effects. The cell concentration at*
15 *the onset of the experiment was 298 cells ml$^{-1}$.* "

Reviewer #2. You started with a very limiting silicate concentration, so how could you state that darkness is responsible for the death of the cells? Multiple nutrient stress in the batch may have be the trigger for the diatom death…
20 *Authors.-* We already modified the manuscript (in the previous revision round) to avoid such confusion. Darkness, low nutrients, the presence of other microorganisms including viruses and parasites, among other threats are the conditions at the aphotic layer of the ocean for phytoplankton. Our experiment, testing cell death of natural populations close to "*in situ*" conditions, represent an original contribution. We added a paragraph in the revised manuscript to
25 improve the discussion and highlight the value of these results. Note, however, that the results obtained in this experiment are consistent with those in previous papers, so it is, therefore, logical to expect that darkness contributed to the mortality observed.

*Action:* In p 8, lines 27-31we added the following paragraph: "*Beyond the stressor of*
30 *continuous darkness, the fast decay rates observed here under aphotic experimental conditions may have been influenced, in addition to darkness, by low nutrients availability, and/or the possible presence of pathogens or parasites, suggesting that survival of natural populations below the photic layer may be lower than expected in the dark from axenic, high- nutrients "in vitro" studies with cultures, aspect already pointed by Ignitiades and Smayda (1970).*

Reviewer #2. The authors already observed some living diatoms really deep in the water column during the Malaspina expedition, so why would it be different in arctic?

*Authors.-* We, again, emphasize that there is no conflict between these observations, as we also report in our paper that between 0.5 % and 50% of diatoms retrieved from the aphotic layer (50 to 400 m) in our Arctic study were living (Malaspina samples were from 2000-4000 m depth, Agusti et al. 2015). Hence, our observations and those from the Malaspina Expedition are
45 consistent.
*Action:* Table 2 shows the percentage of aphotic living cells to vary between 0.5 and 50 %.

Reviewer #2.  Diatom sinking triggered by the health status of the cells:
As there was more than one sampling, it could have been nice at least to have the composition of the sample at the different time. However, the authors need to be a lot more careful in their discussion as they only tested one community once. While the idea behind is interesting, it is not

5 enough experiment to conclude.
*Authors.-*  We agree that it would have been nice to conduct more experiments.  However, since there was ample prior evidence that dead cells sink faster (elegantlty demonstrated by Smayda, 1971) we believe that a single experiment, showing that these findings also apply to the Arctic community tested, sufficed.

10 *Action:*  The experiment was not designed to assess how the community may have changed inside the experimental chamber (although we reasonabluy expect no growth under the dark conditions in the chamber. p 7, second paragraph: *"The experiment conducted, although limited in number, was tested in a natural community and showed that dead cells sank much faster than living ones in a field assemblage."*

P5
Line 28: Fig. 2 should be Fig. 3
Fig. 2 was cited properly.

20 Line 36: Navicula does not dominate the Barent Sea the composition look much alike St3
*Action:*  We revised this statement.  The sentence was modified to avoid confusion, Navicula was found at the polar front, and did not dominate the community.  The sentence now reads: "with a larger contribution of other taxa, including *Navicula pelagica* (station #9, Fig. 4)."
P6

25 Line 20: what was the main species after one hour?
*Action:*  -P6 was revised.  After one hour, Coscinodiscus was dominant at the bottom, as already indicated.

Line 30 : how to concile with result showing some Thalassiosira starting back in culture after 70

30 days in the dark?

- We added a new paragraph (in pg 7-lines 22-30) to discuss the literature on the topic. Careful reading of the paper the reviewer extracts this statement from shows that there is indeed not conflict between our results and those observations: *Thalassiosira* vegetative

35 cells died, some vegetative cells, however, led to resting stages were that formed after several days in darkness and the resulting cysts were able to grow when reillumined after 70 days in darkness.

P7Line 29: "depleted silicic acid silicic acid limits diatom Si uptake to such a degree that growth

40 must slow, Krause et al" please add a coma or a colon for clarity
Line 30: but you show that they stay alive only shortly after they leave the photic zone
Line 31-33: I don't understand your sentence here that state that diatom mortality is triggered by Si limitation, especially as what you explained in the next paragraph is more like Si limitation trigger the loss of buoyancy and the darkness trigger the death of the cell, no?

45 *Authors:*  The reviewer is correct that the narrative may lead to confusion. The term mortality referring to the consequences of acute Si limitation was misleading, as we referred to high mortality rate, as reflected in 30-50% of the diatom cells in Si-depleted photic layers to be dead.

The rest are likely to be in poor condition.  Available evidence suggests that this leads to loss of buoyancy capacity that characterized healthy cells, leading to rapid sinking of the bloom.  A fraction of the cells falling below the photic layer is still alive, as shown in Table 1 (0.5 to 50%), but most of these will die under the aphotic conditions.   Overall we feel that the discrepancies with the reviewer stem from confusion between mortality, interpreted by the reviewer to represent the death of 100%, which is consistent with the data we show that characterizes both populations in the photic and aphotic layer as consisting of living and dead cells in variable proportions (i.e. dead or alive is a binary state for individual cells, but a probability for the population).  Hence we refer to mortality rates throughout the paper.

*Action:*  p. 8,  lines 22- 25: We revised and corrected the text to avoid confusion:  " *A large fraction (30 – 50%) of the diatom cells in silicon-depleted photic layer were dead, pointing at acute silicic acid limitation as the likely factor triggering partial mortality, while the remaining cells are likely to be senescent.  Unhealthy diatoms would then lose the capacity to actively regulate buoyancy that characterizes healthy diatom cells (Smayda, 1970), leading to rapid sinking of the bloom.* "
and: " *The potential for rapid sedimentation is enhanced, due to the higher silica quotas for polar diatoms (Lomas et al. 2019) compared to lower-latitude diatoms (Brzezinski 1985).* "

P8
Line 10: aphotic instead of photic?
*Action:*  We have changed this in the revised manuscript.

New references

[revised manuscript text omitted]

**Formatted** … [170]
**Formatted** … [172]
**Formatted** … [173]
**Formatted** … [175]
**Formatted** … [176]
**Formatted** … [177]
**Formatted** … [178]
**Formatted** … [180]
**Formatted** … [181]
**Formatted** … [182]
**Formatted** … [183]
**Formatted** … [184]
**Formatted** … [186]
**Formatted** … [187]
**Formatted** … [188]
**Formatted** … [189]
**Formatted** … [191]
**Formatted** … [193]
**Formatted** … [195]
**Formatted** … [196]
**Formatted** … [197]
**Formatted** … [198]
**Formatted** … [199]
**Formatted** … [201]
**Formatted** … [202]
**Formatted** … [203]
**Formatted** … [204]
**Formatted** … [206]

**Table 1.** Stations number and location, averaged (±SE) photic layer temperature, salinity, upper mixed layer (UPM) depth, nutrients, and measurements made with the Bottle-Net (BN) in the photic and aphotic zones, indicating the depth of the tows, and the abundance and percentage of living diatoms found at the two layers.

| Station | Latitude N | Longitude E | Temperature (°C) | Salinity (psu) | UPM (m) | $NO_3+NO_2$ (µM) | $PO_4$ (µM) | $Si(OH)_4$ (µM) | BN Photic (range, m) | BN Aphotic (range, m) | Photic diatoms (cells m$^{-2}$) | Aphotic diatoms (cells m$^{-2}$) | Photic living diatoms (%) | Aphotic living diatoms (%) |
|---|---|---|---|---|---|---|---|---|---|---|---|---|---|---|
| # 3, Bellsund Hula | 77 28.09 | 13 27.483 | 0.81 ± 0.33 | 34.48 ± 0.083 | 14.5 | 1.79 ± 1.52 | 0.27 ± 0.11 | 0.75 ± 0.45 | 45-0 | 197-55 | 3.160E+07 | 6.843E+06 | 63.54 | 21.70 |
| # 4, Bredjupet | 77 03.356 | 13 23.369 | 4.64 ± 0.025 | 35.0 ± 0.001 | 65.5 | 9.44 ± 0.097 | 0.63 ± 0.019 | 4.16 ± 0.046 | 60-0 | 415-100 | 3.04E+05 | 1.63E+05 | 20.93 | 9.47 |
| # 5, Inngang Hornsund | 76 58.73 | 15 44.113 | -0.54 ± 0.035 | 34.27 ± 0.037 | 24.2 | 5.66 ± 0.019 | 0.34 ± 0.078 | 2.45 ± 0.40 | 50-0 | 220-80 | 3.20E+07 | 3.00E+05 | 72.03 | 0.50 |
| # 6, Hornsund Dypet | 76 51.244 | 15 13.143 | -0.034 ± 0.1 | 28.98 ± 4.5 | 9.8 | 0.49 ± 0.37 | 0.17 ± 0.03 | 0.36 ± 0.118 | 50-0 | 220-60 | 2.01E+09 | 4.69E+08 | 70.03 | 8.31 |
| # 7, Erik Eriksen Strait | 79 09.986 | 26 02.20 | -1.44 ± 0.093 | 34.29 ± 0.04 | 35.9 | 0.03 ± 0.026 | 0.16 ± 0.01 | 0.07 ± 0.012 | 50-0 | 260-70 | 1.25E+07 | 1.13E+06 | 61.12 | 26.99 |
| # 8, Erik Eriksen Strait | 79 10.479 | 26 27.518 | -1.31 ± 0.088 | 34.22 ± 0.4 | 3.0 | 2.23 ± 1.64 | 0.15 ± 0.077 | 0.57 ± 0.40 | 50-0 | 245-70 | 2.47E+10 | 5.56E+06 | 69.79 | 31.27 |
| # 9, Polar Front | 77 15.308 | 29 29.243 | 2.04 ± 0.099 | 34.7 ± 0.027 | 34.0 | 0.14 ± 0.034 | 0.204 ± 0.022 | 1.29 ± 0.17 | 50-0 | 180-60 | 2.76E+07 | 2.27E+06 | 45.97 | 50.00 |
| # 10, Barents Sea | 76 13.513 | 29 43.710 | 4.06 ± 0.044 | 34.9 ± 0.001 | 75.0 | 3.21 ± 0.20 | 0.345 ± 0.03 | 1.48 ± 0.156 | 50-0 | 180-60 | 1.45E+08 | 2.35E+07 | 71.77 | 13.14 |

[Figure]

**Figure 1**

[Figure]

**Figure 2**

[Figure]

5 **Figure 3**

[Figure]

**Figure 4**

(a)

[Figure]

(b)

[Figure]

**Figure 5**

[Figure]

**Figure 6**

[Figure]

[Figure]

**Figure 7**

| Page 10: [1] Commented [JK2] | Jeffrey Krause | 5/6/19 10:43:00 AM |

Lomas, M.W., Baer, S.E., Acton, S., Krause, J.W.  2019.  Pumped up by the Cold:  Elemental Quotas and Stoichiometry of Polar Diatoms. *Frontiers in Marine Science*.  doi: 10.3389/fmars.2019.00197

| Page 10: [2] Commented [JK3] | Jeffrey Krause | 5/6/19 10:43:00 AM |

Baines, Stephen B., Benjamin S. Twining, Mark A. Brzezinski, David M. Nelson, and Nicholas S. Fisher. "Causes and biogeochemical implications of regional differences in silicification of marine diatoms." *Global Biogeochemical Cycles* 24, no. 4 (2010).

| Page 11: [3] Deleted | Jeffrey Krause | 5/6/19 10:37:00 AM |

| Page 11: [3] Deleted | Jeffrey Krause | 5/6/19 10:37:00 AM |

| Page 11: [3] Deleted | Jeffrey Krause | 5/6/19 10:37:00 AM |

| Page 11: [3] Deleted | Jeffrey Krause | 5/6/19 10:37:00 AM |

| Page 11: [3] Deleted | Jeffrey Krause | 5/6/19 10:37:00 AM |

| Page 11: [4] Deleted | Jeffrey Krause | 5/6/19 10:38:00 AM |

| Page 11: [4] Deleted | Jeffrey Krause | 5/6/19 10:38:00 AM |

| Page 11: [4] Deleted | Jeffrey Krause | 5/6/19 10:38:00 AM |

| Page 11: [4] Deleted | Jeffrey Krause | 5/6/19 10:38:00 AM |

| Page 11: [4] Deleted | Jeffrey Krause | 5/6/19 10:38:00 AM |

| Page 11: [4] Deleted | Jeffrey Krause | 5/6/19 10:38:00 AM |

| Page 11: [4] Deleted | Jeffrey Krause | 5/6/19 10:38:00 AM |

| Page 11: [4] Deleted | Jeffrey Krause | 5/6/19 10:38:00 AM |

| Page 11: [4] Deleted | Jeffrey Krause | 5/6/19 10:38:00 AM |
|---|---|---|

| Page 11: [5] Formatted | Jeffrey Krause | 5/6/19 12:53:00 PM |
|---|---|---|

English (UK)

| Page 11: [5] Formatted | Jeffrey Krause | 5/6/19 12:53:00 PM |
|---|---|---|

English (UK)

| Page 11: [6] Formatted | Jeffrey Krause | 5/6/19 12:53:00 PM |
|---|---|---|

Subscript

| Page 11: [6] Formatted | Jeffrey Krause | 5/6/19 12:53:00 PM |
|---|---|---|

Subscript

| Page 11: [7] Deleted | Jeffrey Krause | 5/6/19 10:47:00 AM |
|---|---|---|

| Page 11: [7] Deleted | Jeffrey Krause | 5/6/19 10:47:00 AM |
|---|---|---|

| Page 11: [8] Formatted | Jeffrey Krause | 5/6/19 12:53:00 PM |
|---|---|---|

English (UK)

| Page 11: [8] Formatted | Jeffrey Krause | 5/6/19 12:53:00 PM |
|---|---|---|

English (UK)

| Page 11: [9] Deleted | Jeffrey Krause | 5/6/19 10:48:00 AM |
|---|---|---|

| Page 11: [10] Formatted | Jeffrey Krause | 5/6/19 12:53:00 PM |
|---|---|---|

English (UK)

| Page 11: [10] Formatted | Jeffrey Krause | 5/6/19 12:53:00 PM |
|---|---|---|

English (UK)

| Page 11: [10] Formatted | Jeffrey Krause | 5/6/19 12:53:00 PM |
|---|---|---|

English (UK)

| Page 11: [11] Deleted | Jeffrey Krause | 5/6/19 10:50:00 AM |
|---|---|---|

| Page 11: [11] Deleted | Jeffrey Krause | 5/6/19 10:50:00 AM |
|---|---|---|

| Page 11: [11] Deleted | Jeffrey Krause | 5/6/19 10:50:00 AM |
|---|---|---|

| Page 11: [12] Formatted | Jeffrey Krause | 5/6/19 12:53:00 PM |
|---|---|---|

English (UK)

| Page 11: [12] Formatted | Jeffrey Krause | 5/6/19 12:53:00 PM |
|---|---|---|

English (UK)

| Page 11: [12] Formatted | Jeffrey Krause | 5/6/19 12:53:00 PM |
|---|---|---|

English (UK)

| Page 11: [13] Deleted | Jeffrey Krause | 5/6/19 10:51:00 AM |
|---|---|---|

| Page 11: [13] Deleted | Jeffrey Krause | 5/6/19 10:51:00 AM |
|---|---|---|

| Page 11: [13] Deleted | Jeffrey Krause | 5/6/19 10:51:00 AM |
|---|---|---|

| Page 11: [14] Formatted | Jeffrey Krause | 5/6/19 12:53:00 PM |
|---|---|---|

English (UK)

| Page 11: [14] Formatted | Jeffrey Krause | 5/6/19 12:53:00 PM |
|---|---|---|

English (UK)

| Page 12: [15] Formatted | Jeffrey Krause | 5/6/19 12:53:00 PM |
|---|---|---|

English (UK)

| Page 12: [16] Formatted | Jeffrey Krause | 5/6/19 12:53:00 PM |
|---|---|---|

English (UK)

| Page 12: [17] Deleted | Jeffrey Krause | 5/6/19 10:53:00 AM |
|---|---|---|

| Page 12: [18] Formatted | Jeffrey Krause | 5/6/19 12:53:00 PM |
|---|---|---|

English (UK)

| Page 12: [19] Deleted | Jeffrey Krause | 5/6/19 10:54:00 AM |
|---|---|---|

| Page 12: [20] Formatted | Jeffrey Krause | 5/6/19 12:53:00 PM |
|---|---|---|

English (UK)

| Page 12: [21] Deleted | Jeffrey Krause | 5/6/19 10:56:00 AM |
|---|---|---|

| Page 12: [21] Deleted | Jeffrey Krause | 5/6/19 10:56:00 AM |
|---|---|---|

| Page 12: [22] Formatted | Jeffrey Krause | 5/6/19 12:53:00 PM |
|---|---|---|

English (UK)

| Page 12: [23] Formatted | Jeffrey Krause | 5/6/19 12:53:00 PM |
|---|---|---|

English (UK)

| Page 12: [24] Deleted | Jeffrey Krause | 5/6/19 10:56:00 AM |
|---|---|---|

| Page 12: [24] Deleted | Jeffrey Krause | 5/6/19 10:56:00 AM |
|---|---|---|

| Page 12: [25] Formatted | Jeffrey Krause | 5/6/19 12:53:00 PM |
|---|---|---|

English (UK)

| Page 12: [25] Formatted | Jeffrey Krause | 5/6/19 12:53:00 PM |
|---|---|---|

English (UK)

| Page 12: [26] Formatted | Jeffrey Krause | 5/6/19 12:49:00 PM |
|---|---|---|

Indent: First line:  1.27 cm

| Page 12: [27] Formatted | Jeffrey Krause | 5/6/19 12:53:00 PM |
|---|---|---|

English (UK)

| Page 12: [28] Formatted | Jeffrey Krause | 5/6/19 12:53:00 PM |
|---|---|---|

English (UK)

| Page 12: [29] Deleted | Jeffrey Krause | 5/6/19 12:40:00 PM |
|---|---|---|

| Page 12: [29] Deleted | Jeffrey Krause | 5/6/19 12:40:00 PM |
|---|---|---|

| Page 12: [30] Formatted | Jeffrey Krause | 5/6/19 12:53:00 PM |
|---|---|---|

English (UK)

| Page 12: [31] Formatted | Jeffrey Krause | 5/6/19 12:53:00 PM |
|---|---|---|

English (UK)

| Page 12: [32] Deleted | Jeffrey Krause | 5/6/19 12:45:00 PM |
|---|---|---|

| Page 12: [32] Deleted | Jeffrey Krause | 5/6/19 12:45:00 PM |
|---|---|---|

| Page 12: [32] Deleted | Jeffrey Krause | 5/6/19 12:45:00 PM |
|---|---|---|

| Page 12: [32] Deleted | Jeffrey Krause | 5/6/19 12:45:00 PM |
|---|---|---|

| Page 12: [32] Deleted | Jeffrey Krause | 5/6/19 12:45:00 PM |
|---|---|---|

| Page 12: [33] Formatted | Jeffrey Krause | 5/6/19 12:53:00 PM |
|---|---|---|

English (UK)

| Page 12: [34] Formatted | Jeffrey Krause | 5/6/19 12:53:00 PM |
|---|---|---|

English (UK)

| Page 12: [35] Formatted | Jeffrey Krause | 5/6/19 12:53:00 PM |

English (UK)

| Page 12: [35] Formatted | Jeffrey Krause | 5/6/19 12:53:00 PM |

English (UK)

| Page 12: [36] Deleted | Jeffrey Krause | 5/6/19 12:45:00 PM |

| Page 12: [36] Deleted | Jeffrey Krause | 5/6/19 12:45:00 PM |

| Page 12: [37] Formatted | Jeffrey Krause | 5/6/19 12:53:00 PM |

English (UK)

| Page 12: [37] Formatted | Jeffrey Krause | 5/6/19 12:53:00 PM |

English (UK)

| Page 12: [38] Deleted | Jeffrey Krause | 5/6/19 12:46:00 PM |

| Page 12: [38] Deleted | Jeffrey Krause | 5/6/19 12:46:00 PM |

| Page 12: [39] Formatted | Jeffrey Krause | 5/6/19 12:53:00 PM |

English (UK)

| Page 12: [39] Formatted | Jeffrey Krause | 5/6/19 12:53:00 PM |

English (UK)

| Page 12: [40] Formatted | Jeffrey Krause | 5/6/19 12:53:00 PM |

English (UK)

| Page 12: [41] Formatted | Jeffrey Krause | 5/6/19 12:53:00 PM |

English (UK)

| Page 12: [41] Formatted | Jeffrey Krause | 5/6/19 12:53:00 PM |

English (UK)

| Page 12: [42] Formatted | Jeffrey Krause | 5/6/19 12:53:00 PM |

English (UK)

| Page 12: [43] Deleted | Jeffrey Krause | 5/6/19 12:48:00 PM |

| Page 12: [44] Formatted | Jeffrey Krause | 5/6/19 12:53:00 PM |

English (UK)

| Page 12: [45] Formatted | Microsoft Office User | 5/12/19 10:37:00 AM |

Font: Not Italic, Complex Script Font: Not Italic

| Page 12: [46] Deleted | Microsoft Office User | 5/12/19 10:37:00 AM |

| Page 12: [47] Formatted | Jeffrey Krause | 5/6/19 12:53:00 PM |
|---|---|---|

English (UK)

| Page 12: [47] Formatted | Jeffrey Krause | 5/6/19 12:53:00 PM |
|---|---|---|

English (UK)

| Page 12: [47] Formatted | Jeffrey Krause | 5/6/19 12:53:00 PM |
|---|---|---|

English (UK)

| Page 12: [48] Formatted | Jeffrey Krause | 5/6/19 12:53:00 PM |
|---|---|---|

English (UK)

| Page 12: [49] Deleted | Jeffrey Krause | 5/6/19 12:51:00 PM |
|---|---|---|

| Page 12: [49] Deleted | Jeffrey Krause | 5/6/19 12:51:00 PM |
|---|---|---|

| Page 13: [50] Formatted | Jeffrey Krause | 5/6/19 12:53:00 PM |
|---|---|---|

English (UK)

| Page 13: [50] Formatted | Jeffrey Krause | 5/6/19 12:53:00 PM |
|---|---|---|

English (UK)

| Page 13: [50] Formatted | Jeffrey Krause | 5/6/19 12:53:00 PM |
|---|---|---|

English (UK)

| Page 13: [50] Formatted | Jeffrey Krause | 5/6/19 12:53:00 PM |
|---|---|---|

English (UK)

| Page 13: [50] Formatted | Jeffrey Krause | 5/6/19 12:53:00 PM |
|---|---|---|

English (UK)

| Page 13: [51] Formatted | Jeffrey Krause | 5/6/19 12:53:00 PM |
|---|---|---|

English (UK)

| Page 13: [51] Formatted | Jeffrey Krause | 5/6/19 12:53:00 PM |
|---|---|---|

English (UK)

| Page 13: [52] Formatted | Jeffrey Krause | 5/6/19 12:53:00 PM |
|---|---|---|

English (UK)

| Page 13: [53] Formatted | Jeffrey Krause | 5/6/19 12:53:00 PM |
|---|---|---|

English (UK)

| Page 13: [54] Formatted | Jeffrey Krause | 5/6/19 12:53:00 PM |
|---|---|---|

English (UK)

| Page 13: [55] Formatted | Jeffrey Krause | 5/6/19 12:53:00 PM |
|---|---|---|

English (UK)

| Page 13: [56] Deleted | Jeffrey Krause | 5/6/19 12:53:00 PM |
|---|---|---|

| Page 13: [56] Deleted | Jeffrey Krause | 5/6/19 12:53:00 PM |

| Page 13: [57] Formatted | Jeffrey Krause | 5/6/19 12:53:00 PM |

English (UK)

| Page 13: [58] Deleted | Jeffrey Krause | 5/6/19 12:54:00 PM |

| Page 13: [58] Deleted | Jeffrey Krause | 5/6/19 12:54:00 PM |

| Page 13: [59] Deleted | Jeffrey Krause | 5/6/19 12:54:00 PM |

| Page 13: [59] Deleted | Jeffrey Krause | 5/6/19 12:54:00 PM |

| Page 13: [59] Deleted | Jeffrey Krause | 5/6/19 12:54:00 PM |

| Page 13: [59] Deleted | Jeffrey Krause | 5/6/19 12:54:00 PM |

| Page 13: [59] Deleted | Jeffrey Krause | 5/6/19 12:54:00 PM |

| Page 13: [59] Deleted | Jeffrey Krause | 5/6/19 12:54:00 PM |

| Page 13: [59] Deleted | Jeffrey Krause | 5/6/19 12:54:00 PM |

| Page 13: [60] Formatted | Jeffrey Krause | 5/6/19 12:53:00 PM |

English (UK)

| Page 13: [61] Deleted | Jeffrey Krause | 5/6/19 12:57:00 PM |

| Page 13: [61] Deleted | Jeffrey Krause | 5/6/19 12:57:00 PM |

| Page 13: [62] Formatted | Jeffrey Krause | 5/6/19 12:53:00 PM |

English (UK)

| Page 13: [62] Formatted | Jeffrey Krause | 5/6/19 12:53:00 PM |

English (UK)

| Page 13: [63] Formatted | Jeffrey Krause | 5/6/19 12:53:00 PM |

English (UK)

**Page 13: [64] Deleted**          **Jeffrey Krause**          **5/6/19 12:57:00 PM**

**Page 13: [65] Formatted**          **Jeffrey Krause**          **5/6/19 12:53:00 PM**

English (UK)

**Page 13: [66] Formatted**          **Jeffrey Krause**          **5/6/19 12:53:00 PM**

English (UK)

**Page 13: [67] Formatted**          **Jeffrey Krause**          **5/6/19 12:53:00 PM**

English (UK)

**Page 13: [68] Formatted**          **Jeffrey Krause**          **5/6/19 12:53:00 PM**

English (UK)

**Page 13: [69] Formatted**          **Jeffrey Krause**          **5/6/19 12:53:00 PM**

English (UK)

**Page 13: [70] Formatted**          **Jeffrey Krause**          **5/6/19 12:53:00 PM**

English (UK)

**Page 13: [71] Deleted**          **Jeffrey Krause**          **5/6/19 12:59:00 PM**

**Page 13: [71] Deleted**          **Jeffrey Krause**          **5/6/19 12:59:00 PM**

**Page 13: [72] Formatted**          **Jeffrey Krause**          **5/6/19 12:53:00 PM**

English (UK)

**Page 13: [72] Formatted**          **Jeffrey Krause**          **5/6/19 12:53:00 PM**

English (UK)

**Page 13: [73] Deleted**          **Jeffrey Krause**          **5/6/19 12:59:00 PM**

**Page 13: [73] Deleted**          **Jeffrey Krause**          **5/6/19 12:59:00 PM**

**Page 13: [73] Deleted**          **Jeffrey Krause**          **5/6/19 12:59:00 PM**

**Page 13: [74] Formatted**          **Jeffrey Krause**          **5/6/19 12:53:00 PM**

English (UK)

**Page 13: [75] Formatted**          **Jeffrey Krause**          **5/6/19 12:53:00 PM**

English (UK)

**Page 13: [76] Formatted**          **Microsoft Office User**          **5/10/19 10:52:00 AM**

Font: 12 pt, Complex Script Font: 12 pt, English (US)

| Page 13: [77] Formatted | Jeffrey Krause | 5/6/19 12:53:00 PM |

English (UK)

| Page 13: [78] Deleted | Jeffrey Krause | 5/6/19 1:08:00 PM |

| Page 14: [79] Formatted | Jeffrey Krause | 5/6/19 12:53:00 PM |

English (UK)

| Page 14: [79] Formatted | Jeffrey Krause | 5/6/19 12:53:00 PM |

English (UK)

| Page 14: [80] Formatted | Jeffrey Krause | 5/6/19 12:53:00 PM |

English (UK)

| Page 14: [81] Deleted | Jeffrey Krause | 5/6/19 1:11:00 PM |

| Page 14: [82] Formatted | Jeffrey Krause | 5/6/19 12:53:00 PM |

English (UK)

| Page 14: [83] Formatted | Jeffrey Krause | 5/6/19 12:53:00 PM |

English (UK)

| Page 14: [83] Formatted | Jeffrey Krause | 5/6/19 12:53:00 PM |

English (UK)

| Page 14: [84] Formatted | Jeffrey Krause | 5/6/19 12:53:00 PM |

English (UK)

| Page 14: [85] Formatted | Jeffrey Krause | 5/6/19 12:53:00 PM |

English (UK)

| Page 14: [85] Formatted | Jeffrey Krause | 5/6/19 12:53:00 PM |

English (UK)

| Page 14: [86] Formatted | Jeffrey Krause | 5/6/19 12:53:00 PM |

English (UK)

| Page 14: [86] Formatted | Jeffrey Krause | 5/6/19 12:53:00 PM |

English (UK)

| Page 14: [87] Formatted | Jeffrey Krause | 5/6/19 12:53:00 PM |

English (UK)

| Page 14: [87] Formatted | Jeffrey Krause | 5/6/19 12:53:00 PM |

English (UK)

| Page 14: [88] Deleted | Susana R. Agusti | 4/19/19 9:31:00 AM |

| Page 14: [88] Deleted | Susana R. Agusti | 4/19/19 9:31:00 AM |

| Page 14: [89] Formatted | Jeffrey Krause | 5/6/19 12:53:00 PM |
|---|---|---|

English (UK)

| Page 14: [90] Deleted | Jeffrey Krause | 5/6/19 1:14:00 PM |
|---|---|---|

| Page 14: [90] Deleted | Jeffrey Krause | 5/6/19 1:14:00 PM |
|---|---|---|

| Page 14: [91] Formatted | Jeffrey Krause | 5/6/19 12:53:00 PM |
|---|---|---|

English (UK)

| Page 14: [92] Formatted | Jeffrey Krause | 5/6/19 12:53:00 PM |
|---|---|---|

English (UK)

| Page 14: [93] Deleted | Jeffrey Krause | 5/6/19 1:15:00 PM |
|---|---|---|

| Page 14: [93] Deleted | Jeffrey Krause | 5/6/19 1:15:00 PM |
|---|---|---|

| Page 14: [94] Formatted | Jeffrey Krause | 5/6/19 12:53:00 PM |
|---|---|---|

English (UK)

| Page 14: [95] Deleted | Jeffrey Krause | 5/6/19 1:15:00 PM |
|---|---|---|

| Page 14: [95] Deleted | Jeffrey Krause | 5/6/19 1:15:00 PM |
|---|---|---|

| Page 14: [95] Deleted | Jeffrey Krause | 5/6/19 1:15:00 PM |
|---|---|---|

| Page 14: [96] Formatted | Jeffrey Krause | 5/6/19 12:53:00 PM |
|---|---|---|

English (UK)

| Page 14: [97] Formatted | Jeffrey Krause | 5/6/19 12:53:00 PM |
|---|---|---|

English (UK)

| Page 14: [97] Formatted | Jeffrey Krause | 5/6/19 12:53:00 PM |
|---|---|---|

English (UK)

| Page 14: [98] Formatted | Jeffrey Krause | 5/6/19 12:53:00 PM |
|---|---|---|

English (UK)

| Page 14: [99] Formatted | Jeffrey Krause | 5/6/19 12:53:00 PM |
|---|---|---|

English (UK)

| Page 14: [99] Formatted | Jeffrey Krause | 5/6/19 12:53:00 PM |
|---|---|---|

English (UK)

| Page 14: [100] Formatted | Jeffrey Krause | 5/6/19 12:53:00 PM |
|---|---|---|

English (UK)

| Page 14: [101] Formatted | Jeffrey Krause | 5/6/19 12:53:00 PM |

English (UK)

| Page 14: [101] Formatted | Jeffrey Krause | 5/6/19 12:53:00 PM |

English (UK)

| Page 14: [102] Formatted | Jeffrey Krause | 5/6/19 12:53:00 PM |

English (UK)

| Page 14: [102] Formatted | Jeffrey Krause | 5/6/19 12:53:00 PM |

English (UK)

| Page 14: [103] Deleted | Jeffrey Krause | 5/6/19 1:19:00 PM |

| Page 14: [103] Deleted | Jeffrey Krause | 5/6/19 1:19:00 PM |

| Page 14: [104] Formatted | Jeffrey Krause | 5/6/19 12:53:00 PM |

English (UK)

| Page 14: [105] Formatted | Jeffrey Krause | 5/6/19 12:53:00 PM |

English (UK)

| Page 14: [106] Deleted | Jeffrey Krause | 5/6/19 1:21:00 PM |

| Page 14: [107] Formatted | Jeffrey Krause | 5/6/19 12:53:00 PM |

English (UK)

| Page 15: [108] Deleted | Jeffrey Krause | 5/6/19 1:21:00 PM |

| Page 15: [109] Formatted | Jeffrey Krause | 5/6/19 12:53:00 PM |

English (UK)

| Page 15: [109] Formatted | Jeffrey Krause | 5/6/19 12:53:00 PM |

English (UK)

| Page 15: [110] Formatted | Jeffrey Krause | 5/6/19 12:53:00 PM |

English (UK), Superscript

| Page 15: [110] Formatted | Jeffrey Krause | 5/6/19 12:53:00 PM |

English (UK), Superscript

| Page 15: [110] Formatted | Jeffrey Krause | 5/6/19 12:53:00 PM |

English (UK), Superscript

| Page 15: [110] Formatted | Jeffrey Krause | 5/6/19 12:53:00 PM |

English (UK), Superscript

| Page 15: [110] Formatted | Jeffrey Krause | 5/6/19 12:53:00 PM |

English (UK), Superscript

| Page 15: [111] Deleted | Susana R. Agusti | 4/12/19 9:48:00 AM |
|---|---|---|

| Page 15: [111] Deleted | Susana R. Agusti | 4/12/19 9:48:00 AM |
|---|---|---|

| Page 15: [111] Deleted | Susana R. Agusti | 4/12/19 9:48:00 AM |
|---|---|---|

| Page 15: [112] Formatted | Jeffrey Krause | 5/6/19 12:53:00 PM |
|---|---|---|

English (UK)

| Page 15: [113] Formatted | Jeffrey Krause | 5/6/19 12:53:00 PM |
|---|---|---|

English (UK)

| Page 15: [114] Formatted | Jeffrey Krause | 5/6/19 12:53:00 PM |
|---|---|---|

English (UK)

| Page 15: [115] Deleted | Microsoft Office User | 5/12/19 10:34:00 AM |
|---|---|---|

| Page 15: [115] Deleted | Microsoft Office User | 5/12/19 10:34:00 AM |
|---|---|---|

| Page 15: [116] Formatted | Jeffrey Krause | 5/6/19 12:53:00 PM |
|---|---|---|

English (UK)

| Page 15: [117] Commented [SRA4] | Susana R. Agusti | 4/12/19 9:57:00 AM |
|---|---|---|

Smayda, T. J., Mitchell-Innes, B.: Dark survival of autotrophic, planktonic marine diatoms. Marine

Biology, 25, 195-202, 1974.

| Page 15: [118] Formatted | Jeffrey Krause | 5/6/19 12:53:00 PM |
|---|---|---|

English (UK)

| Page 15: [118] Formatted | Jeffrey Krause | 5/6/19 12:53:00 PM |
|---|---|---|

English (UK)

| Page 15: [118] Formatted | Jeffrey Krause | 5/6/19 12:53:00 PM |
|---|---|---|

English (UK)

| Page 15: [118] Formatted | Jeffrey Krause | 5/6/19 12:53:00 PM |
|---|---|---|

English (UK)

| Page 15: [119] Formatted | Jeffrey Krause | 5/6/19 12:53:00 PM |
|---|---|---|

English (UK)

| Page 15: [120] Formatted | Jeffrey Krause | 5/6/19 12:53:00 PM |
|---|---|---|

English (UK)

**Page 15: [121] Commented [SRA5]**    Susana R. Agusti    4/20/19 11:51:00 AM

Segovia, M. and Berges, J. A. (2009), INHIBITION OF CASPASE-LIKE ACTIVITIES PREVENTS THE APPEARANCE OF REACTIVE OXYGEN SPECIES AND DARK-INDUCED APOPTOSIS IN THE UNICELLULAR CHLOROPHYTE *DUNALIELLA TERTIOLECTA*1. Journal of Phycology, 45: 1116-1126. doi:10.1111/j.1529-8817.2009.00733.x
* * *
**Page 15: [122] Formatted**    Jeffrey Krause    5/6/19 12:53:00 PM

English (UK)

**Page 15: [122] Formatted**    Jeffrey Krause    5/6/19 12:53:00 PM

English (UK)

**Page 15: [123] Formatted**    Jeffrey Krause    5/6/19 12:53:00 PM

English (UK)

**Page 15: [123] Formatted**    Jeffrey Krause    5/6/19 12:53:00 PM

English (UK)

**Page 15: [124] Commented [SRA6]**    Susana R. Agusti    4/20/19 12:08:00 PM

Cell Death in the Unicellular Chlorophyte *Dunaliella tertiolecta*. A Hypothesis on the Evolution of Apoptosis in Higher Plants and Metazoans
María Segovia, Liti Haramaty, John A. Berges, Paul G. Falkowski
Plant Physiology May 2003, 132 (1) 99-105; **DOI:** 10.1104/pp.102.017129
* * *
**Page 15: [125] Formatted**    Jeffrey Krause    5/6/19 12:53:00 PM

English (UK)

**Page 15: [125] Formatted**    Jeffrey Krause    5/6/19 12:53:00 PM

English (UK)

**Page 15: [126] Formatted**    Jeffrey Krause    5/6/19 12:53:00 PM

English (UK)

**Page 15: [126] Formatted**    Jeffrey Krause    5/6/19 12:53:00 PM

English (UK)

**Page 15: [127] Commented [SRA7]**    Susana R. Agusti    4/27/19 10:39:00 AM

Ignatiades, L., & Smayda, T. J. (1970). AUTECOLOGICAL STUDIES ON THE MARINE DIATOM RHIZOSOLENIA FRAGILISSIMA BERGON. II. ENRICHMENT AND DARK VIABILITY EXPERIMENTS 1. *Journal of Phycology*, *6*(4), 357-364.
* * *
**Page 15: [128] Formatted**    Jeffrey Krause    5/6/19 12:53:00 PM

English (UK)

**Page 15: [128] Formatted**    Jeffrey Krause    5/6/19 12:53:00 PM

English (UK)

**Page 15: [129] Formatted**    Jeffrey Krause    5/6/19 2:20:00 PM

Indent: First line:  1.27 cm

**Page 15: [130] Formatted**    Jeffrey Krause    5/6/19 12:53:00 PM

English (UK)

| Page 15: [131] Formatted | Jeffrey Krause | 5/6/19 12:53:00 PM |

English (UK)

| Page 15: [131] Formatted | Jeffrey Krause | 5/6/19 12:53:00 PM |

English (UK)

| Page 15: [132] Formatted | Microsoft Office User | 5/12/19 1:54:00 PM |

Font: Not Italic, Complex Script Font: Italic

| Page 15: [132] Formatted | Microsoft Office User | 5/12/19 1:54:00 PM |

Font: Not Italic, Complex Script Font: Italic

| Page 15: [133] Commented [JK8] | Jeffrey Krause | 5/6/19 1:39:00 PM |

Susana,
See if you agree with this assessment (next sentences).

| Page 15: [134] Formatted | Microsoft Office User | 5/12/19 1:54:00 PM |

Complex Script Font: Italic

| Page 15: [135] Formatted | Jeffrey Krause | 5/6/19 2:01:00 PM |

Subscript

| Page 16: [136] Deleted | Microsoft Office User | 5/12/19 1:53:00 PM |

| Page 16: [136] Deleted | Microsoft Office User | 5/12/19 1:53:00 PM |

| Page 16: [137] Formatted | Jeffrey Krause | 5/6/19 12:53:00 PM |

English (UK)

| Page 16: [137] Formatted | Jeffrey Krause | 5/6/19 12:53:00 PM |

English (UK)

| Page 16: [138] Deleted | Jeffrey Krause | 5/6/19 2:19:00 PM |

| Page 16: [139] Formatted | Jeffrey Krause | 5/6/19 12:53:00 PM |

English (UK)

| Page 16: [140] Deleted | Jeffrey Krause | 5/6/19 2:31:00 PM |

| Page 16: [141] Formatted | Jeffrey Krause | 5/6/19 12:53:00 PM |

English (UK)

| Page 16: [142] Formatted | Jeffrey Krause | 5/6/19 12:53:00 PM |

English (UK)

| Page 16: [142] Formatted | Jeffrey Krause | 5/6/19 12:53:00 PM |

English (UK)

| Page 16: [143] Formatted | Jeffrey Krause | 5/6/19 12:53:00 PM |
|---|---|---|

English (UK)

| Page 16: [143] Formatted | Jeffrey Krause | 5/6/19 12:53:00 PM |
|---|---|---|

English (UK)

| Page 16: [144] Deleted | Jeffrey Krause | 5/6/19 2:32:00 PM |
|---|---|---|

| Page 16: [144] Deleted | Jeffrey Krause | 5/6/19 2:32:00 PM |
|---|---|---|

| Page 16: [145] Formatted | Jeffrey Krause | 5/6/19 12:53:00 PM |
|---|---|---|

English (UK)

| Page 16: [146] Deleted | Jeffrey Krause | 5/6/19 2:32:00 PM |
|---|---|---|

| Page 16: [146] Deleted | Jeffrey Krause | 5/6/19 2:32:00 PM |
|---|---|---|

| Page 16: [147] Formatted | Jeffrey Krause | 5/6/19 12:53:00 PM |
|---|---|---|

English (UK)

| Page 16: [147] Formatted | Jeffrey Krause | 5/6/19 12:53:00 PM |
|---|---|---|

English (UK)

| Page 16: [148] Formatted | Jeffrey Krause | 5/6/19 12:53:00 PM |
|---|---|---|

English (UK)

| Page 16: [148] Formatted | Jeffrey Krause | 5/6/19 12:53:00 PM |
|---|---|---|

English (UK)

| Page 16: [149] Deleted | Jeffrey Krause | 5/6/19 2:33:00 PM |
|---|---|---|

| Page 16: [149] Deleted | Jeffrey Krause | 5/6/19 2:33:00 PM |
|---|---|---|

| Page 16: [149] Deleted | Jeffrey Krause | 5/6/19 2:33:00 PM |
|---|---|---|

| Page 16: [150] Deleted | Jeffrey Krause | 5/6/19 2:33:00 PM |
|---|---|---|

| Page 16: [151] Formatted | Jeffrey Krause | 5/6/19 12:53:00 PM |
|---|---|---|

English (UK)

| Page 16: [152] Formatted | Jeffrey Krause | 5/6/19 12:53:00 PM |
|---|---|---|

English (UK)

| Page 16: [153] Formatted | Jeffrey Krause | 5/6/19 12:53:00 PM |
|---|---|---|

English (UK)

| Page 16: [154] Formatted | Jeffrey Krause | 5/6/19 12:53:00 PM |
|---|---|---|

English (UK)

| Page 16: [155] Formatted | Jeffrey Krause | 5/6/19 12:53:00 PM |
|---|---|---|

English (UK)

| Page 16: [156] Formatted | Jeffrey Krause | 5/6/19 12:53:00 PM |
|---|---|---|

English (UK)

| Page 16: [157] Formatted | Jeffrey Krause | 5/6/19 12:53:00 PM |
|---|---|---|

English (UK)

| Page 16: [158] Formatted | Jeffrey Krause | 5/6/19 12:53:00 PM |
|---|---|---|

English (UK)

| Page 16: [159] Formatted | Jeffrey Krause | 5/6/19 12:53:00 PM |
|---|---|---|

English (UK)

| Page 16: [160] Deleted | Jeffrey Krause | 5/6/19 2:35:00 PM |
|---|---|---|

| Page 16: [160] Deleted | Jeffrey Krause | 5/6/19 2:35:00 PM |
|---|---|---|

| Page 16: [160] Deleted | Jeffrey Krause | 5/6/19 2:35:00 PM |
|---|---|---|

| Page 16: [161] Formatted | Jeffrey Krause | 5/6/19 12:53:00 PM |
|---|---|---|

English (UK)

| Page 16: [162] Deleted | Jeffrey Krause | 5/6/19 2:37:00 PM |
|---|---|---|

| Page 16: [162] Deleted | Jeffrey Krause | 5/6/19 2:37:00 PM |
|---|---|---|

| Page 16: [162] Deleted | Jeffrey Krause | 5/6/19 2:37:00 PM |
|---|---|---|

| Page 16: [163] Formatted | Jeffrey Krause | 5/6/19 12:53:00 PM |
|---|---|---|

English (UK)

| Page 16: [164] Formatted | Jeffrey Krause | 5/6/19 12:53:00 PM |
|---|---|---|

English (UK)

| Page 16: [165] Formatted | Microsoft Office User | 5/12/19 2:18:00 PM |
|---|---|---|

Font: Not Italic, Complex Script Font: Not Italic

| Page 16: [166] Formatted | Jeffrey Krause | 5/6/19 12:53:00 PM |
|---|---|---|

English (UK)

**Page 16: [167] Formatted**      **Jeffrey Krause**      **5/6/19 12:53:00 PM**

English (UK)

**Page 16: [168] Commented [JK13]**      **Jeffrey Krause**      **5/6/19 2:39:00 PM**

Brzezinski, Mark A. "The Si: C: N ratio of marine diatoms: Interspecific variability and the effect of some environmental variables1." *Journal of Phycology* 21, no. 3 (1985): 347-357.

**Page 16: [169] Formatted**      **Jeffrey Krause**      **5/6/19 12:53:00 PM**

English (UK)

**Page 22: [170] Formatted**      **Jeffrey Krause**      **5/6/19 12:53:00 PM**

English (UK)

**Page 22: [171] Deleted**      **Jeffrey Krause**      **5/6/19 3:03:00 PM**

**Page 22: [171] Deleted**      **Jeffrey Krause**      **5/6/19 3:03:00 PM**

**Page 22: [172] Formatted**      **Jeffrey Krause**      **5/6/19 12:53:00 PM**

English (UK)

**Page 22: [173] Formatted**      **Jeffrey Krause**      **5/6/19 12:53:00 PM**

English (UK)

**Page 22: [173] Formatted**      **Jeffrey Krause**      **5/6/19 12:53:00 PM**

English (UK)

**Page 22: [174] Deleted**      **Jeffrey Krause**      **5/6/19 3:04:00 PM**

**Page 22: [174] Deleted**      **Jeffrey Krause**      **5/6/19 3:04:00 PM**

**Page 22: [175] Formatted**      **Jeffrey Krause**      **5/6/19 12:53:00 PM**

English (UK)

**Page 22: [176] Formatted**      **Jeffrey Krause**      **5/6/19 12:53:00 PM**

English (UK)

**Page 22: [177] Formatted**      **Jeffrey Krause**      **5/6/19 12:53:00 PM**

English (UK)

**Page 22: [178] Formatted**      **Jeffrey Krause**      **5/6/19 12:53:00 PM**

English (UK)

**Page 22: [179] Deleted**      **Jeffrey Krause**      **5/6/19 3:18:00 PM**

**Page 22: [179] Deleted**      **Jeffrey Krause**      **5/6/19 3:18:00 PM**

| Page 22: [180] Formatted | Jeffrey Krause | 5/6/19 12:53:00 PM |
|---|---|---|

English (UK)

| Page 22: [180] Formatted | Jeffrey Krause | 5/6/19 12:53:00 PM |
|---|---|---|

English (UK)

| Page 22: [181] Formatted | Jeffrey Krause | 5/6/19 12:53:00 PM |
|---|---|---|

English (UK)

| Page 22: [182] Formatted | Jeffrey Krause | 5/6/19 12:53:00 PM |
|---|---|---|

English (UK)

| Page 22: [183] Formatted | Jeffrey Krause | 5/6/19 12:53:00 PM |
|---|---|---|

English (UK)

| Page 22: [183] Formatted | Jeffrey Krause | 5/6/19 12:53:00 PM |
|---|---|---|

English (UK)

| Page 22: [184] Formatted | Jeffrey Krause | 5/6/19 12:53:00 PM |
|---|---|---|

English (UK)

| Page 22: [185] Deleted | Jeffrey Krause | 5/6/19 3:20:00 PM |
|---|---|---|

| Page 22: [185] Deleted | Jeffrey Krause | 5/6/19 3:20:00 PM |
|---|---|---|

| Page 22: [186] Formatted | Jeffrey Krause | 5/6/19 12:53:00 PM |
|---|---|---|

English (UK)

| Page 22: [187] Formatted | Jeffrey Krause | 5/6/19 12:53:00 PM |
|---|---|---|

English (UK)

| Page 22: [187] Formatted | Jeffrey Krause | 5/6/19 12:53:00 PM |
|---|---|---|

English (UK)

| Page 22: [188] Formatted | Jeffrey Krause | 5/6/19 12:53:00 PM |
|---|---|---|

English (UK)

| Page 22: [188] Formatted | Jeffrey Krause | 5/6/19 12:53:00 PM |
|---|---|---|

English (UK)

| Page 22: [189] Formatted | Jeffrey Krause | 5/6/19 12:53:00 PM |
|---|---|---|

English (UK)

| Page 22: [189] Formatted | Jeffrey Krause | 5/6/19 12:53:00 PM |
|---|---|---|

English (UK)

| Page 22: [190] Deleted | Jeffrey Krause | 5/6/19 3:07:00 PM |
|---|---|---|

| Page 22: [190] Deleted | Jeffrey Krause | 5/6/19 3:07:00 PM |
|---|---|---|

**Page 22: [191] Formatted**       **Jeffrey Krause**       **5/6/19 12:53:00 PM**

English (UK)

**Page 22: [191] Formatted**       **Jeffrey Krause**       **5/6/19 12:53:00 PM**

English (UK)

**Page 22: [192] Deleted**       **Jeffrey Krause**       **5/6/19 3:07:00 PM**

**Page 22: [192] Deleted**       **Jeffrey Krause**       **5/6/19 3:07:00 PM**

**Page 22: [192] Deleted**       **Jeffrey Krause**       **5/6/19 3:07:00 PM**

**Page 22: [193] Formatted**       **Jeffrey Krause**       **5/6/19 12:53:00 PM**

English (UK)

**Page 22: [194] Deleted**       **Jeffrey Krause**       **5/6/19 3:22:00 PM**

**Page 22: [194] Deleted**       **Jeffrey Krause**       **5/6/19 3:22:00 PM**

**Page 22: [195] Formatted**       **Jeffrey Krause**       **5/6/19 12:53:00 PM**

English (UK)

**Page 22: [196] Formatted**       **Jeffrey Krause**       **5/6/19 12:53:00 PM**

English (UK)

**Page 22: [196] Formatted**       **Jeffrey Krause**       **5/6/19 12:53:00 PM**

English (UK)

**Page 22: [197] Formatted**       **Jeffrey Krause**       **5/6/19 12:53:00 PM**

English (UK)

**Page 22: [198] Formatted**       **Jeffrey Krause**       **5/6/19 12:53:00 PM**

English (UK)

**Page 22: [199] Formatted**       **Jeffrey Krause**       **5/6/19 12:53:00 PM**

English (UK)

**Page 22: [200] Deleted**       **Jeffrey Krause**       **5/6/19 3:10:00 PM**

**Page 22: [200] Deleted**       **Jeffrey Krause**       **5/6/19 3:10:00 PM**

**Page 22: [201] Formatted**       **Jeffrey Krause**       **5/6/19 12:53:00 PM**

English (UK)

**Page 22: [202] Formatted**       **Jeffrey Krause**       **5/6/19 12:53:00 PM**

English (UK)

| Page 22: [203] Formatted | Jeffrey Krause | 5/6/19 12:53:00 PM |

English (UK)

| Page 22: [204] Formatted | Jeffrey Krause | 5/6/19 12:53:00 PM |

English (UK)

| Page 22: [205] Deleted | Jeffrey Krause | 5/6/19 3:11:00 PM |

| Page 22: [205] Deleted | Jeffrey Krause | 5/6/19 3:11:00 PM |

| Page 22: [206] Formatted | Jeffrey Krause | 5/6/19 12:53:00 PM |

English (UK)

| Page 22: [207] Formatted | Jeffrey Krause | 5/6/19 12:53:00 PM |

English (UK)

| Page 22: [207] Formatted | Jeffrey Krause | 5/6/19 12:53:00 PM |

English (UK)

| Page 22: [208] Deleted | Jeffrey Krause | 5/6/19 3:25:00 PM |

| Page 22: [208] Deleted | Jeffrey Krause | 5/6/19 3:25:00 PM |

---

## Author Response (AR3)

**Actions taken to accommodate the comments of reviewer #3 on "Arctic (Svalbard Islands) Active and Exported Diatom  Stocks and Cell Health Status"** by Susana Agustí et al.
https://doi.org/10.5194/bg-2018-459-RC1, 2018

Reviewer #3.- The article studies the impact of life stage of diatoms on their vertical export in the Svalbard area. The article would confirm the hypothesis that unhealthy diatoms are sinking whereas the healthy ones maintain better buoyancy and rather stay at surface.
I quite enjoyed the reading and I found it very instructive.
First, the topic is highly strategic and interesting. Diatoms dominate the primary production in this very productive region. However, very few (or no?) studies have in situ measurements. The hypothesis that senescent diatoms sink is often use without proper in situ proof. For these reasons, I think this article could potentially provide a valuable contribution.
However, I could just agree with the other reviewers. I have concerns about the significativity of the results which are based on a very limited dataset. Yes, the authors found significative differences between photic and aphotic communities but still, this is a very weak. I acknowledge that the authors have done a great effort to re-frame their results during the review process and that there is not much more information to extract from the data set. I recommend say 'moderate' revisions to address remaining issues.

*Authors:* We thank the reviewer for the useful comments, which help us improve the manuscript further. We have revised the manuscript following all of your recommendations as detailed below.

Reviewer #3.- About the rate of mortality, I agree with most of the authors answer to reviewer #2. But I deeply encourage authors to be more specific. For example, it was not clear, before I read the answer to reviewer #2 the subtilty in the terminology of "survival". Please clarify that you only consider vegetative cells here. Because this is clear there is survival of (resting) cells in the dark for a long period of time. Perhaps it could serve also in the discussion to understand that your approach is different from other studies. I have to admit I had exactly the same reaction than reviewer #2 during my first read of the revised manuscript (first appeared contradictory with the literature I know)

*Authors:* We agree that such clarifications are needed to avoid confussion, as happened in earlier version, and have revised the text and clarified this aspect.
*Action:*  In lines 22-25, p7, we added the sentence: "*The decay rates calculated for living or viable vegetative cells were more than three times faster than those observed for the total cells in the population. These included both viable and non-viable vegetative cells, which are, however, morphologically similar and could not be differentiated unless using specific methods to discriminate living from dead cells, such as the staining test used here.*"

Reviewer #3.- I was also surprised some major references were not listed in your study. As global critic, this article still lacks of contextualization. Some articles (see below in gray) that I found related (you are not obliged to cite those but it could help you to improve the discussion).
See for example:
Kvernvik et al. 2019, https://doi.org/10.1111/jpy.12750. They say for example: "Our results suggest that some Arctic autotrophs maintain fully functional photosystem II and downstream electron acceptors during the polar night... This could allow Arctic microalgae to endure the polar night without the formation of dormant stages, enabling them to recover and take advantage of light immediately upon the suns return during the winter–spring transition.". I am really surprised the authors did not take more precaution while discussing this still debated topic.
In Berge et al. 2015:
"Many Arctic phototrophic plankters are able to persist during unfavorable conditions as resting stages such as spores or cysts (Garrison, 1984; Smetacek, 1985; Krempand Anderson, 2000), and diatoms are known for their potential to survive long periods of darkness (Antia and Cheng, 1970; Smayda and Mitchell-Innes, 1974; Palmisano and Sullivan, 1982; Sakshaug et al., 2009; Quillfeldt et al., 2009). The survival strategies of the various plastidic flagellates of Arctic waters throughout the dark period, however, are largely unknown."
Lacour et al. (2019) also suggest the opposite (https://link.springer.com/article/10.1007%2Fs00300-019-02507-2))
"Chaetoceros neogracilis was not able to grow in the dark but cell biovolume remained constant after 1 month in darkness. Rapid resumption of photosynthesis and growth recovery was also found when the

cells were transferred back to light at four different light levels ranging from 5 to 154 μmol photon m−2 s−1. This demonstrates the remarkable ability of this species to re-initiate growth over a wide range of irradiances even after a prolonged period in the dark with no apparent lag period or impact on survival." As well as the extensive synthesis by Wulf et al. 2008 (https://www.tandfonline.com/doi/abs/10.1080/0269249X.2008.9705774) "Based on the rapid increase in Fv/Fm we safely draw the conclusion that although the cells probably were physiologically resting in the dark they were not forming resting stagessuch as spores or cysts. Physiologically resting cells are morphologically similar to the vegetative cells, but are physiologically dormant and can be induced when cells are transferred to cold and dark conditions (Anderson 1975a). Like in our study, these cells have condensed protoplasts which are transformed back to the former state (within hours) upon re-exposure"

*Authors:*  We agree with the reviewer that additional contextualization based on published research will improve the paper and thank you for the references suggested.  Indeed, the papers by Berge et al. 2015, Kvernvik et al. 2019, along with other studies, are very relevant. Most existing research, including the papers recommended, focus on explaining or testing the capacity of winter phytoplankton to resist darkness, which is relevant, but differs from our experiment goals and sampling conditions. Our experiment was run in the spring time with shallow mixing and no night, as solar radiation was received 24h per day.  The phytoplankton we sampled during the cruise was not acclimated/selected to resist darkness, and most probably was photo-acclimated to high-light or long-photoperiod conditions. Hence, the results expected are quite different from those expected if the experiments had been done with phytoplankton sampled in the Arctic winter (i.e. prolonged darkness) as done in many of the papers suggested or existing literature with cultures.

*Action:*  We added a new paragraph to the revised discussion where we updated the results reported in the context of new references, including the references indicated by the reviewer.  The new paragraph (p7-8) reads: "*Our results reporting fast diatom cell death under aphotic conditions are contrasting with the expectation of high survival capacity of polar diatoms to darkness supported by existing evidence. Recent reports identified fast photosynthetic response to irradiance in diatoms sampled during the dark winter time around the Svalbard Islands (Kvernvik et al., 2019).  Phenotypic selection for specific physiological properties allows polar diatoms to acclimate to low light and darkness (Lacour et al 2019).  Our experiment, however, tested vegetative cell survival in a spring community under a 24 h light: 0 hours darkness photoperiod and with shallow mixing depths. The community tested, was, therefore, likely photo-acclimated to prolonged photoperiod and relatively high irradiance.  Therefore, it is expected to respond differently under darkness than arctic microalgae growing underneath the ice or under very short photoperiods and/or minimal irradiance levels (Lacour et al., 2019; Berge et al., 2015). Hence, Arctic phytoplankton is expected to show contrasting responses to prolonged darkness in the spring, when acclimated to long photoperiods, than in winter, where surviving cells are expected to be acclimated to short photoperiods.*"

Reviewer#3-  How the authors can be 100% confident that their unique culture experiment was reliable? Did something else than darkness could have killed the algae (i.e. contamination?) Could you discuss that ?

*Authors:*  We agree that additional experiments would add more confidence on our findings, where the experiment discussed is but one supporting element, for a central data set consisting of diatom stocks and viability in situ. We hope that our experiment, which obviously elicits a lot of interest (as indicated by the comments of this and previous reviewers focusing on this experiment), will encourage the community to conduct additional experiments.
*Action:*  We added the following sentence to the revised manuscript (Lines 11-13, p8): "*We cannot rule out that contamination from the vessel during experimental preparations may affected mortality rates in the incubations. As the decline rates were derived from a single test, further experiments on cell decay rates of Arctic diatoms sampled in the spring under dark conditions will be required to confirm our results*".

Reviewer#3- Also, there is many syntax, grammar and terminology issues in the text (lot of them were introduced during the revision process, that is a pity). I recommend to carefully correct the text because it

makes the understanding sometimes difficult (I had to re-read many sentences many times). I have tried to list them in the specific comments but it became quickly verwhelming.

I encourage the authors to continue their efforts. Specific comments:
PAGE9 (abstract).
L.23: specify phytoplankton bloom, it could be sea ice algae. Or if no distinction, just microalgae.
L.26: regional = specify Svalbard or Northwest Barents Sea
L.27-28: very awkward sentence. Change "conccurrent with" something like "together with".
L.29: SE= Standard Error ? I don't think you could use undefined abbreviation in the abstract. Need confirmation from editor.
PAGE11 (2.1)
L.23: you canno't say MLD is an indicator of stability. Stability is usually related to stratification. Stratification is basically the density gradient which you don't measure here. MLD is an ndicator of mixing, this is it. People know what is MLD (or UPM), I suggest you just erase "an index of the stability of the water column" and also later in the text.
PAGE12 (2.3)
L.32 Move "expected" before "mortality".
L.35: I can understand but this is not well written. Perhaps change ", this simulated" par "simulating" ? or "which simulated" ?
PAGE13 (3)
please be consistent and use either r or R2 throughout the manuscript.
PAGE14
L.10: N or n ?
L.13: "from station 6 to station8"
L.14: a E is missing in the verb were. Change "these wre also the areas with" by something like "and where"
L.31-32: this is interpretation, should be moved in discussion.
PAGE 15, LINE 5 and PAGE 16 LINE 7: please do not use the term trend in this context. This is not appropriate, please re-word.
PAGE16
Line 23: amoung out ??? amoung OUR ?
Line 35-37: weird wording, had to re-read several times. I guess you wanted to say sedimentation is enhanced BY higher quotas for polar diatoms. Please re-word.

*Action:* We thank the reviewer for all the edits and suggestions. We have revised the wording, and corrected all the specific comments in the revised manuscript.

[revised manuscript text omitted]

5    **Figure 3**

[Figure]

**Figure 4**

(a)

[Figure]

(b)

[Figure]

**Figure 5**

[Figure]

**Figure 6**

[Figure]

**Figure 7**

| Page 12: [1] Formatted | Susana | 8/28/19 9:19:00 AM |
|---|---|---|

Font: Not Italic, Complex Script Font: Not Italic, English (US)

| Page 12: [2] Formatted | Susana | 8/28/19 9:19:00 AM |
|---|---|---|

Font: Not Italic, Complex Script Font: Not Italic

| Page 12: [3] Formatted | Susana | 8/28/19 7:11:00 PM |
|---|---|---|

Font: Not Italic, Complex Script Font: Not Italic, English (US)